# Metabolic Dysregulation and Its Role in Postoperative Pain among Knee Osteoarthritis Patients

**DOI:** 10.3390/ijms25073857

**Published:** 2024-03-29

**Authors:** Elena V. Tchetina, Kseniya E. Glemba, Galina A. Markova, Svetlana I. Glukhova, Maksim A. Makarov, Aleksandr M. Lila

**Affiliations:** 1Immunology and Molecular Biology Department, Nasonova Research Institute of Rheumatology, Moscow 115522, Russia; 2Surgery Department, Nasonova Research Institute of Rheumatology, Moscow 115522, Russiaortopedniir@mail.ru (M.A.M.); 3Statistics Department, Nasonova Research Institute of Rheumatology, Moscow 115522, Russia; 4Osteoartritis Laboratory, Nasonova Research Institute of Rheumatology, Moscow 115522, Russia; amlila@mail.ru

**Keywords:** knee osteoarthritis, postoperative pain, metabolic dysregulation, arthroplasty outcomes, carbohydrate and fatty acid metabolism, gene expression, peripheral blood

## Abstract

Knee osteoarthritis (KOA) is characterized by low-grade inflammation, loss of articular cartilage, subchondral bone remodeling, synovitis, osteophyte formation, and pain. Strong, continuous pain may indicate the need for joint replacement in patients with end-stage OA, although postoperative pain (POP) of at least a two-month duration persists in 10–40% of patients with OA. Study purpose: The inflammation observed in joint tissues is linked to pain caused by the production of proinflammatory cytokines. Since the biosynthesis of cytokines requires energy, their production is supported by extensive metabolic conversions of carbohydrates and fatty acids, which could lead to a disruption in cellular homeostasis. Therefore, this study aimed to investigate the association between POP development and disturbances in energy metabolic conversions, focusing on carbohydrate and fatty acid metabolism. Methods: Peripheral blood samples were collected from 26 healthy subjects and 50 patients with end-stage OA before joint replacement surgery. All implants were validated by orthopedic surgeons, and patients with OA demonstrated no inherent abnormalities to cause pain from other reasons than OA disease, such as malalignment, aseptic loosening, or excessive bleeding. Pain levels were assessed before surgery using the visual analogue scale (VAS) and neuropathic pain questionnaires, DN4 and PainDETECT. Functional activity was evaluated using the Western Ontario and McMaster Universities Osteoarthritis Index (WOMAC). Three and six months after surgery, pain indices according to a VAS of 30 mm or higher were considered. Total RNA isolated from whole blood was analyzed using quantitative real-time RT-PCR (qRT-PCR) for the expression of genes related to carbohydrate and fatty acid metabolism. Protein levels of the examined genes were measured using an ELISA in the peripheral blood mononuclear cells (PBMCs). We used qRT-PCR because it is the most sensitive and reliable method for gene expression analysis, while an ELISA was used to confirm our qRT-PCR results. Key findings: Among the study cohort, 17 patients who reported POP demonstrated significantly higher (*p* < 0.05) expressions of the genes PKM2, LDH, SDH, UCP2, CPT1A, and ACLY compared to pain-free patients with KOA. Receiver-operating characteristic (ROC) curve analyses confirmed the association between these gene expressions and pain development post-arthroplasty. A principle component analysis identified the prognostic values of ACLY, CPT1A, AMPK, SDHB, Caspase 3, and IL-1β gene expressions for POP development in the examined subjects. Conclusion: These findings suggest that the disturbances in energy metabolism, as observed in the PBMCs of patients with end-stage KOA before arthroplasty, may contribute to POP development. An understanding of these metabolic processes could provide insights into the pathogenesis of KOA. Additionally, our findings can be used in a clinical setting to predict POP development in end-stage patients with KOA before arthroplasty.

## 1. Introduction

Knee osteoarthritis (KOA) is characterized by the loss of articular cartilage, remodeling of the subchondral bone, synovial inflammation, and osteophyte formation [1,2]. Currently, OA affects 6% of the world’s population, with complaints of joint pain discomfort in one-third of subjects over the age of sixty-five [3,4]. Pain is the primary clinical symptom that limits patients’ work and daily self-care capacities, and severe persistent pain becomes a crucial indicator for joint replacement at the late stage of KOA. The prevalent pain therapy for end-stage KOA patients is knee arthroplasty, with a growing number of cases worldwide [5]. However, pain persists in 10–40% of patients after knee replacement surgery [6]. The specific criteria for chronic long-term postoperative pain (POP) were defined recently [7]: the pain must develop after a surgical procedure; the pain must have a duration of at least two months; other causes for the pain must be excluded; and the possibility that the pain is from a pre-existing condition must also be excluded. Chronic POP can last a considerable amount of time, as it has been observed in patients up to 12 years after surgery [8]. Understanding the factors influencing the outcome of arthroplasty related to POP could provide KOA patients with a more realistic understanding of surgical results and optimize their preoperative condition [9].

Local articular cartilage degradation, changes in subchondral bone, and synovial inflammation have long been considered the major characteristics of KOA [1,2]. However, recently, several systemic molecular mechanisms have been implicated in the pathogenesis and progression of the disease. These mechanisms involve chronic low-grade inflammation due to the upregulation of proinflammatory cytokines, which promote the upregulation of matrix metalloproteinases, aggrecanases, and cathepsins, leading to the breakdown of the cartilage extracellular matrix. These activities are accompanied by dysregulation of metabolic pathways, such as glucose and lipid metabolism, and mitochondrial dysfunction, which can affect chondrocyte homeostasis and contribute to cartilage degeneration [10].

Pain in OA is primarily associated with the destruction of joint tissues due to inflammation and the induction of hyperalgesia. At the molecular level, pain is triggered by excessive production of cytokines in chondrocytes, such as tumor necrosis factor (TNF)α, interleukin (IL)-1β, IL6, and nerve growth factor [11]. For instance, recently we have shown that the high expression of genes associated with inflammation, extracellular matrix degradation, and apoptotic activity in the peripheral blood may serve as important biomarkers for the POP development in end-stage patients with KOA [12].

The proliferation of immune cells and the synthesis of proinflammatory cytokines constitute energy-intensive processes associated with the low-grade systemic inflammation observed in OA [13]. In the inflamed tissues, cellular biosynthetic activities require glucose breakdown through aerobic glycolysis to convert pyruvate into lactate (Warburg effect) involving pyruvate kinase 2 (PKM2) and lactate dehydrogenase (LDH) [14]. Indeed, increased activity of PKM2 was previously demonstrated in patients with OA compared to healthy individuals [15]. Simultaneously, oxidative mitochondrial metabolism involving Krebs cycle enzyme activities such as succinate dehydrogenase (SDH), malate dehydrogenase (MDH), and isocitrate dehydrogenase (IDH), as well as the electron transport chain (ETC), associated with anti-inflammatory outcomes, was suppressed in these cells [16,17]. In view of this, the uncoupling of oxidation and phosphorylation, involving a specific uncoupling protein (UCP2), which facilitates proton leakage through the inner mitochondrial membrane and reduces free radical production, also regulates Krebs cycle activity [18].

Furthermore, overweight and obesity are major risk factors of KOA disease development [19], with higher body mass index (BMI) levels correlating with increased pain in OA [20]. Lipid metabolism involves fatty acid catabolic conversions that produce energy in the form of adenosine triphosphate (ATP) [21] and anabolic processes that generate building blocks for biosynthetic purposes [22]. Free long-chain fatty acids enter cells through specific transporters [23] and are converted into Acyl-coenzyme A (Acyl-CoA), which is transferred into the mitochondrial matrix by palmitoyl-CoA transferase (CPT). Subsequent beta-oxidation of Acyl-CoA generates Acetyl-CoA, which condenses with oxaloacetate to enter the Krebs cycle as citrate [21].

Moreover, de novo synthesis of fatty acids from Acetyl-CoA occurs outside the mitochondria, in the cytosol. To produce cytosolic Acetyl-CoA, citrate is transported across the inner mitochondrial membrane [24], where it is cleaved by ATP citrate lyase (ACLY) into Acetyl-CoA and oxaloacetate [25]. Cytosolic Acetyl-CoA is carboxylated by Acetyl-CoA carboxylase (ACACa) to malonyl-CoA, which is the first step in fatty acid synthesis catalyzed by fatty acid synthase (FASN) [26]. The de novo fatty acid synthesis process is regulated by ACACa and malonyl-CoA decarboxylase (MLYCD), which converts malonyl-CoA into Acetyl-CoA and carbon dioxide [27]. 

Previous studies investigating pain in KOA primarily focused on inflammatory mechanisms. However, analyzing proinflammatory cytokine production alone may not uncover the underlying causes of pain development. Therefore, we chose to explore another aspect of cellular vitality related to general energy production mechanisms as a potential primary factor influencing metabolic fluctuations. We hypothesized that POP development in KOA might be associated with overall disturbances in carbohydrate and fatty acid metabolism that can be monitored in peripheral blood mononuclear cells (PBMCs). 

The purpose of the present study was to determine the differences in the expression of genes associated with carbohydrate and fatty acid metabolism in the blood of end-stage patients with knee osteoarthritis before arthroplasty, aiming to determine the molecular mechanisms responsible for the development of chronic POP.

In light of the issues mentioned above, we employed various approaches to analyze gene and protein expressions associated with energy production in the peripheral blood of end-stage KOA patients before surgery. These approaches included correlation studies, ROC curve analyses, logistic regression modeling, principal component analyses, and construction of protein–protein interaction networks to identify and validate genes with significant prognostic power.

## 2. Results

### 2.1. Clinical Parameters of the Examined Patients with End-Stage KOA before Surgery

The demographic and clinical characteristics of 50 patients with end-stage KOA, both male and female, were previously described in detail [9]. In summary, the K&L OA grades of these subjects ranged from III to IV (grade III: 37 patients; grade IV: 13 patients). The average age of these patients was 67.6 years (range: 54–82 years), and the average disease duration was 9.9 years (range: 3–30 years). The majority of patients demonstrated an elevated body mass index (BMI) with an average of 30.5. The average total WOMAC score was 1065, with the total pain score at 222.8, total physical function score at 746, and total stiffness score at 93.9. The Western Ontario and McMaster Universities Arthritis Index (WOMAC) is widely used in the evaluation of KOA. It is a self-administered questionnaire consisting of 24 items divided into three subscales: pain, stiffness, and physical function.

All patients reported persistent knee joint pain over the last three months. Pain assessments using the visual analog scale (VAS) revealed moderate levels of knee pain, with an average score of 64.4 mm (range 20–90 mm). The Brief Pain Inventory (BPI) questionnaire indicated an average pain severity of 4.8. The evaluation of neuropathic pain using the DN4 questionnaire yielded an average score of 1.9. The PainDETECT questionnaire showed an average score of 6.0. According to the Hospital Anxiety and Depression Scale (HADS), 3 out of 50 patients exhibited abnormal levels of anxiety, while 10 patients had borderline anxiety levels. Additionally, 8 out of 50 patients were found to be depressed, and 14 patients had borderline depression scores according to the HADS depression scale. Patients were interviewed regarding pain intensity at 3 and 6 months post-surgery. Post-surgical pain complaints with VAS score (≥30 mm) were reported by 17 out of 50 patients (34%); 33 patients were pain-free at both time points.

The clinical assessment of patients from both subgroups prior to surgery revealed no significant differences in the expression of most examined variables. However, it was noted that in patients who developed POP, arterial hypertension was recorded significantly more often, in 65% of patients compared to 30% of patients who were satisfied with the surgery outcome. Additionally, the following trends were observed: a higher incidence of grade 1 obesity (*p* = 0.08) and elevated anxiety levels, as measured by the HADS questionnaire (*p* = 0.07), were observed in patients who developed postoperative pain. Conversely, the severity of preoperative pain was slightly higher in patients who were satisfied with the surgery results (*p* = 0.07) [12].

However, these differences in the manifestation of the examined clinical parameters of the end-stage KOA patients could not help to distinguish patients who will develop POP or not. In addition, clinical traits cannot provide any indications on the intrinsic mechanisms of POP development. Therefore, we used molecular approaches aiming to solve the problem indicated as the purpose of our study.

### 2.2. Whole-Blood Gene Expression 

Examination of gene expression in the whole blood of the examined patients with end-stage OA revealed a significant upregulation of PKM2 (*p* = 0.03), LDH (*p* = 0.04), SDH (*p* = 0.02), UCP2 (*p* = 0.02), AMPKα (*p* = 0.001), CPT1A (*p* = 0.01), and ACLY (*p* = 0.01) in those who developed post-surgical pain compared to 33 pain-free subjects (Figure 1, Table 1). However, the gene expression of IDH, MDH, PDH, ACACa, MLYCD, FASN, and ATP-synthase (ATP5B) was not significantly different (*p* > 0.05) in both subgroups of end-stage patients with knee OA.

Figure 1 consists of two panels. The left panel presents the expressions of genes related to glycolysis (PKM2 and LDH), the Krebs cycle (SDH), electron transport chain uncoupling protein (UCP2), ATP synthase (ATP5B), and AMP-activated protein kinase (AMPKα), the major regulator of glucose and fatty acid metabolism [28]. The right panel introduces the expressions of genes related to long-chain fatty acid β-oxidation (CPT1A) and fatty acid synthesis (ACLY, ACACa, MLYCD, and FASN) [29]. The numerical data of gene expression analyses are presented in Table 1.

### 2.3. Protein Levels of the Examined Genes in Isolated PBMCs

To assess the clinical significance of the relative expression of the examined genes in the PBMCs of the subjects with end-stage KOA, we analyzed the protein concentrations of PKM2, MDH, ACACa, and ACLY in the PBMC fraction. The protein levels of PKM2 (*p* = 0.03) and ACLY (*p* = 0.02) in the examined seventeen end-stage patients with KOA who developed post-surgical pain were significantly higher compared to pain-free subjects, while MDH and ACACa protein concentrations were not significantly different (*p* > 0.05) between patients from the examined subgroups (Figure 2).

In Figure 2, we demonstrate that the obtained gene expression results align with protein expression data obtained after analyzing protein samples from PBMCs of the same patients using an ELISA. These analyses revealed significant differences in gene expressions of PKM2 and ACLY between the examined subgroups of end-stage KOA patients, supported by significant differences in protein concentrations within the same subgroups. Conversely, the lack of significant differences in the gene expression of MDH and ACACa resulted in no significant difference in their cellular protein levels.

The comparison of outcomes from analyses of clinical and molecular profiles suggests that gene expression examination is a powerful tool for differentiating patients prior to arthroplasty. This approach not only helps identify patients at risk of postoperative pain but also provides insights into the potential reasons for POP development. 

### 2.4. Correlation Analyses of the Gene Expressions with Clinical and Radiographic Parameters

Bivariate correlation analyses revealed a positive correlation between BMI and UCP2 and MLYCD gene expressions (Table 2). Neuropathic-pain-related DN4 scores showed positive correlation with ACLY and CPTA1 gene expressions. AMPKα and MLYCD gene expressions also positively correlated with PainDETECT scores. Conversely, BPI pain severity exhibited a negative correlation with MDH gene expression. Additionally, several non-statistically significant trends were observed in relation to the expressions mentioned above and the examined clinical indices.

The correlation analyses revealed that neuropathic pain, as indicated by DN4 and PainDETECT scores, is linked to the overall regulation of cellular energy metabolism (AMPKα), glycolysis (PKM2), respiration efficacy (UCP2), and both the beta-oxidation of long-chain fatty acids (CPT1A) and their de novo synthesis (ACLY and MLYCD).

### 2.5. Validation of Prognostic Values of the Identified Gene Expressions

The prognostic values of these gene expressions were assessed using ROC curve analyses (Figure 3), which confirmed a potential diagnostic/predictive value of expressions of the examined genes before surgery with the likelihood of pain development after surgery. The cut-off values for the examined gene expressions were 7.74 for PKM2 [AUC = 0.689, 95%CI (0.532–0.846), *p* = 0.031, sensitivity of 0.70% and specificity of 0.63%], 5.37 for LDH [AUC = 0.700, 95%CI (0.545–0.856), *p* = 0.023, sensitivity of 0.63% and specificity of 0.65%], 20.06 for AMPKα [AUC = 0.777, 95%CI (0.605–0.950)], *p* = 0.002, sensitivity of 0.75% and specificity of 0.88%], 3.03 for UCP2 [AUC = 0.716, 95%CI (0.546–0.885), *p* = 0.015, sensitivity of 0.69% and specificity of 0.79%], 5.24 for SDH [AUC = 0.758, 95%CI (0.597–0.919), *p* = 0.004, sensitivity of 0.69% and specificity of 0.75%,], 16.01 for CPT1A [AUC = 0.719, 95%CI (0.588–0.773), *p* = 0.02, sensitivity of 0.59% and specificity of 0.77%], and 11.92 for ACLY [AUC = 0.727, 95%CI (0.563–0.891), *p* = 0.01, sensitivity of 0.65% and specificity of 0.82%].

The above-mentioned cut-off values obtained in the ROC curve analyses allow for the use of gene expression data measured in peripheral blood in end-stage KOA patients for predicting postoperative pain (POP) before surgery. Specifically, gene expression values exceeding the cut-off value indicate a higher likelihood of POP development.

The reliability of these cut-off values was confirmed by logistic regression modeling, which showed that high expressions of AMPKα (*p* = 0.0025), SDH (*p* = 0.03), CPT1A (*p* = 0.03), and ACLY (*p* = 0.028) were independent predictors of POP development. However, these results warrant further investigation due to the limited number of patients in both subsets.

### 2.6. Protein–Protein Interaction (PPI) Network Construction

Factor analysis, specifically principle component analysis (PCA), was utilized to reduce the number of identified predictors of POP based on gene expression studies in peripheral blood before surgery in patients with end-stage knee OA from the present study (n = 14) and our previous study (n = 8): cathepsins S (CTSS) and K (CTSK), Caspase 3, TNFα, IL-1β, TIMP1 (tissue inhibitor of metalloproteinase 1), MMP-9 (metalloproteinase-9), and COX2 (cyclooxygenase 2) [9]. Through the PCA, the predominant prognostic values of ACLY, CPT1A, AMPKα, SDH, Caspase 3, and IL-1β gene expressions were identified in patients who developed POP (Figure 4A). Assessing these gene expressions in the peripheral blood may be a suitable approach for predicting POP development in end-stage patients with KOA prior to surgery. Furthermore, the PCA results supported the prognostic significance of the same gene expressions identified through ROC curve analyses and logistic regression modeling. Nevertheless, the PCA of gene expressions in pain-free KOA patients revealed a greater number of interconnected genes, indicating stronger regulatory mechanisms associated with less severe disease (Figure 4B).

The PPI network analysis of the data obtained using the STRING database aimed to integrate the identified associations between proteins. Protein–protein interactions (PPIs) represent specific physical contacts between two or more protein molecules driven by biochemical events involving electrostatic forces, hydrogen bonding, and hydrophobic effects. The degree of connectivity in PPI networks is determined by the number of links to a given node. In the present study, the PPI network analysis showed that end-stage patients who developed POP exhibited a strong association between ACLY and SDH genes, which are linked to fatty acid synthesis and respiratory activities, respectively (Figure 5A). At the same time, the degree of connectivity and the number of genes involved were stronger in patients who were satisfied with surgery results, indicating more powerful regulatory connections between various metabolic pathways in these subjects (Figure 5B).

## 3. Discussion

The weak inflammatory response in OA necessitates a substantial expenditure of cellular energy [30], leading to the emerging concept of an interaction between cellular bioenergetics/metabolism and inflammation [31]. Indeed, excessive proinflammatory cytokine synthesis in inflamed tissues requires glucose breakdown through aerobic glycolysis to convert pyruvate into lactate (known as the Warburg effect) [14]. This mechanism allows for the rapid generation of high amounts of energy in the form of ATP [32]. Simultaneously, the anti-inflammatory activities of oxidative mitochondrial metabolism are suppressed [17]. In healthy subjects, acute inflammation is followed by the restoration of energy balance with a predominance of Krebs cycle activity [33]. However, during chronic inflammation in KOA patients, these mechanisms could be dysregulated.

Actually, in this study, we found that the expression of genes linked to glucose oxidation and fatty acid metabolism was significantly elevated in all the examined patients with KOA. This finding aligns with the increased expression of the proinflammatory cytokines TNFα and IL-1β observed in our previous study within the same patient cohort [12] and with metabolomics studies indicating Krebs cycle dysfunction in OA patients [34].

Mechanistically, inflammation fueled by the upregulation of glycolysis leads to the release of inflammatory mediators, which can bind to their respective receptors on nociceptive neurons in the periphery, modulating their sensitivity and excitability, resulting in increased pain sensitivity [35]. Indeed, we observed that patients with KOA who developed POP exhibited significantly higher expression of glycolysis-related genes in blood cells compared to patients who were satisfied with the surgery results. This observation is supported by animal studies of early OA showing metabolic dysfunction at the level of LDH expression [36]. However, no significant differences were noted in the expressions of genes associated with Krebs cycle activity (specifically IDH and MDH) between patients in both subgroups. The absence of significant changes in the expression of genes related to oxidative mitochondrial metabolism support previous observations of Krebs cycle downregulation compared to glycolysis activity associated with inflammation [17].

However, SDH gene expression was significantly higher in patients with KOA who developed POP, consistent with reports of SDH dysregulation in early OA in mice and humans [37]. The upregulation of SDH has also been associated with proinflammatory immune cell activation [38] and reactive oxygen species (ROS) overproduction [39]. Therefore, the increased expression of the UCP2 gene in patients who developed POP compared to pain-free patients is plausible. Additionally, higher expression of the AMPKα gene in patients who developed POP suggests greater ATP energy requirements compared to pain-free patients, likely linked to the excessive synthesis of proinflammatory cytokines.

It is worth noting that, although animal studies support molecular findings in humans, end-stage KOA disease affects older patients, making it challenging to accurately model in animal studies. Additionally, OA disease in animals is artificially induced by known agents, while the molecular triggers of human OA are currently unclear. Therefore, results from animal model studies should always be verified by human research.

The molecular mechanisms underlying the involvement of fatty acid conversions in POP development in patients with KOA remain unclear. However, the imbalance in intracellular fatty acid composition contributes to increased proinflammatory lipid levels, primarily in adipose tissue, with increased pain and knee joint dysfunction [40]. Therefore, the significant increase in the expression of examined genes associated with fatty acid metabolism in the blood of both subgroups of the end-stage patients with KOA compared to healthy individuals implies an association between fatty acid metabolism and the clinical manifestations of the disease.

Interestingly, the expression of the FASN gene, responsible for de novo fatty acid synthesis, was similar in both examined groups of OA patients and healthy subjects. This observation may be associated with the significantly higher expression of the MLYCD gene, which prevents further polymerization of malonyl-CoA, along with high ACACa gene expression. Therefore, mild inflammation in OA may result from the lack of excessive synthesis of long-chain fatty acids de novo, which are required for cell membrane synthesis in proliferating cells [41]. This observation aligns with the inhibition of long-chain fatty acid synthesis during the suppression of Th17 cell differentiation leading to improved disease condition in animals with collagen-induced arthritis [42].

On the other hand, POP development in patients with end-stage KOA may be associated with a higher degree of fatty acid metabolism imbalance, caused by significantly greater CPT1A and ACLY expressions, potentially leading to increased Acetyl-CoA production by the blood cells compared to patients satisfied with surgery results. This condition increases the likelihood of a non-enzymatic, uncontrolled acetylation of protein molecules [43], resulting in an increased amount of acetylated fraction [44] and functional disturbances [45].

Furthermore, the association of gene expression related to glucose and fatty acid metabolism with POP development is supported by a positive correlation of AMPKα, PKM2, UCP2, ACLY, MLYCD, and CPT1A gene expressions with neuropathic pain scores in end-stage KOA patients’ blood, which suggests the possible involvement of neurogenic mechanisms in POP development [46]. The exacerbation of neuropathic pain indices with an increase in the expression of these genes is supported by a positive correlation of DN4 scores with the expression of proinflammatory cytokines TNFα and IL-1β reported previously [12,47].

Overall, our study demonstrated that chronic POP is associated with significant metabolic dysfunction involving carbohydrate and fatty acid metabolism in end-stage patients with KOA. This is further supported by recent observations showing extensive genetic correlations between metabolic syndrome and OA, revealing a shared genetic architecture, pleiotropic loci, and causality [48]. As pain in the knee joint is considered a key trait related to OA disease, we can suggest that similar metabolic disturbances may occur in patients with OA at other disease stages. Moreover, as we monitored these changes in peripheral blood, metabolic dysfunction may involve other tissues and organs of the body. Therefore, OA might not be a disease exclusive to the joint but rather a systemic condition involving the entire body, and knee arthroplasty may not represent the end stage of OA disease.

Therefore, our findings suggest new approaches for POP diagnostics and treatment. These approaches can enhance the comprehensive evaluation of preoperative clinical characteristics in end-stage patients with KOA [49]. We have identified a number of prognostic biomarkers for POP development that can be measured in peripheral blood before arthroplasty. This means that end-stage KOA patients at risk of POP development may receive treatment before surgery to target the improvement of their metabolic condition. Additionally, understanding individual metabolic disturbances could be considered as an additional component of ERAS (enhanced recovery after surgery) to prevent post-surgical complications in end-stage patients with KOA [50]. In the future, new approaches for potential therapies could be applied to prevent POP development in end-stage KOA patients by addressing intrinsic disturbances in cellular metabolism.

Our study has several strengths. First, we carefully selected the patients with OA who did not have inherent abnormalities that could cause pain from reasons other than OA, such as malalignment, aseptic loosening, or excessive bleeding. Additionally, patients with vascular insufficiency, bleeding, or thrombophlebitis were excluded from the study as previously indicated in our report within the same patient cohort [12]. Furthermore, intravenous tranexamic acid was administered before the skin incision to prevent postoperative bleeding after arthroplasty [12,51]. Second, all implants were validated by orthopedic surgeons. Third, to explore and support our hypothesis, we utilized various research approaches, including the examination of gene expression and cellular protein levels, correlation and ROC curve analyses, PCA, and PPI network construction. All these approaches yielded similar results. However, our study also had a limitation in that it involved a small number of patients. This limitation may have contributed to non-significant results in some of our tests.

Based on our findings, we propose several areas for future research. For example, further investigation into other metabolic pathways that may be dysregulated in osteoarthritis could provide a more comprehensive understanding of the molecular mechanisms underlying POP development. Additionally, developing targeted therapies based on the identified metabolic disturbances in OA could lead to more effective treatments for the disease prior to surgery. Furthermore, validation studies involving larger patient cohorts are needed to confirm the findings of our study and ensure the reproducibility of the results. Overall, these future research directions have the potential to advance our understanding of the molecular mechanisms underlying osteoarthritis and pave the way for the development of novel therapeutic strategies to improve patient care.

Our study brings several novel contributions to the field of OA and postoperative pain research. We have identified specific metabolic disturbances associated with KOA that have not been extensively studied before. By uncovering dysregulation in key metabolic pathways, our study sheds light on the underlying molecular mechanisms contributing to KOA pathogenesis. Additionally, our findings establish a link between metabolic disturbances and postoperative pain outcomes in KOA patients. This novel insight highlights the potential role of metabolism in modulating pain sensitivity and underscores the importance of considering metabolic factors in pain management strategies for KOA patients. By pinpointing metabolic pathways that are dysregulated in KOA, our study has significant implications for the development of personalized treatment approaches that target specific metabolic pathways to alleviate symptoms and improve patient outcomes. Overall, our study contributes to advancing the understanding of KOA by elucidating the intricate interplay between metabolism and pain.

Our findings not only contribute to the understanding of OA at the molecular level but also have broader implications for public health and socioeconomic factors. As OA is a leading cause of disability worldwide, affecting millions of individuals and placing a significant burden on healthcare systems, by uncovering metabolic disturbances associated with KOA and their link to POP outcomes, our study provides valuable insights that can inform public health strategies for the accurate management of OA. The identification of dysregulation in energy metabolic pathways in KOA opens up possibilities for personalized medicine approaches tailored to individual patients’ metabolic profiles. This precision medicine approach has the potential to optimize treatment outcomes, reduce healthcare costs, and improve patient satisfaction and quality of life. KOA is not only a major health issue but also carries significant socioeconomic implications due to its impact on work productivity, healthcare expenditures, and quality of life. By shedding light on the metabolic factors influencing pain outcomes in KOA patients, our study may lead to more cost-effective treatment strategies and improved outcomes, thereby reducing the economic burden associated with OA. Finally, our study sets the stage for further research into the role of metabolism in OA pathogenesis and pain modulation. Future studies could explore the potential of pursuing specific metabolic pathways for therapeutic interventions, as well as investigate the impact of lifestyle factors, such as diet and exercise, on metabolic disturbances in OA. By placing our findings in the larger context of OA research, public health implications, and socioeconomic impacts, we underscore the significance of our work in advancing knowledge, improving patient care, and addressing the multifaceted challenges posed by this prevalent musculoskeletal condition.

## 4. Materials and Methods

### 4.1. Clinical Testing

Patient demographics, including age and sex, were recorded. The radiographic grade of OA was determined by analyzing weight-bearing anteroposterior radiographs of the knees and counted according to Kellgren and Lawrence [52]. The Western Ontario and McMaster Universities Osteoarthritis Index (WOMAC) visual analogue scale was used to evaluate pain, stiffness, and physical function [53]. Nociceptive pain was evaluated using the visual analogue scale (VAS), whereas neuropathic pain was measured using the PainDETECT [54] and DN4 (Douleur Neuropathique en 4 Questions) [55] questionnaires. The Brief Pain Inventory (BPI) questionnaire was used to assess pain severity [56]. The Hospital Anxiety and Depression Scale (HADS) was used to reveal the levels of anxiety and depression among patients with OA [57].

### 4.2. Quantification of Protein Levels by the Enzyme-Linked Immunosorbent Assay (ELISA)

We collected 10 mL of peripheral blood in vacutainers that contained ethylenediaminetetraacetic acid (EDTA) (Sigma-Aldrich, Inc., St. Louis, MO, USA) to prevent coagulation, ensuring the integrity of the blood samples for subsequent analyses. The blood samples were obtained in a standard manner between 07:00 AM and 09:00 AM after patients fasted overnight and before breakfast. Whole blood was separated using a Ficoll density gradient. PBMCs, which have a lower density than Ficoll-Hypaque (1.077 g/L), were separated from higher-density granulocytes and red blood cells through centrifugation after overlaying the diluted blood on the Ficoll-Hypaque layer. Following centrifugation, the blood samples were divided into plasma enriched with thrombocytes, PBMCs located in the interphase, and a pellet containing granulocytes above the red blood cells. The PBMCs from the interphase were isolated and washed twice in phosphate-buffered saline. Erythrocytes were lysed using a hypotonic buffer (1.6 mM EDTA, 10 mM KHCO_3_, 153 mM NH_4_Cl, pH 7.4) at a 3:1 volume ratio. The isolated PBMCs were then frozen and stored at −80 °C until protein extraction [58].

Concentrations of PKM2 (SEA588Hu), MDH2 (SEH672Hu), and ACACa (SEB28Hu) (Cloud-Clone Corp., Wuhan, China) and of ACLY (E01A1192) (BlueGene Biotech, Shanghai, China) were measured in isolated PBMCs using commercially available enzyme-linked immunosorbent assay (ELISA) kits following the manufacturer’s instructions. Standard solutions were provided in the kit. Diluted samples and standards were added to the appropriate wells of the pre-coated 96-well plate. After incubating for 1 h at 37 °C, a detection reagent was added, followed by another 1 h incubation at 37 °C, a washing procedure, and additional incubation with a second detection reagent for 30 min at 37 °C. Following another washing process, the plate was incubated with a substrate solution for 30 min at 37 °C; then, a stop solution was added, and the plate was immediately read using a microplate reader at 450 nm. The results were expressed per µg of DNA measured in PBMC lysates. PBMC lysates were obtained using Cell Extraction Buffer containing 10 mM Tris, pH 7.4, 100 mM NaCl, 1 mM EDTA, 1 mM EGTA, 1 mM NaF, 20 mM Na_4_P_2_O_7_, 20 mM Na_3_VO_4_, 1% Triton X-100, 10% glycerol, 0.1% SDS, and 0.5% deoxycholate (Invitrogen, Camarillo, CA, USA) supplemented with Protease Inhibitor Cocktail (Sigma-Aldrich, Inc., St. Louis, MO, USA) and 1 mM PMSF (Sigma-Aldrich, Inc., St. Louis, MO, USA) according to the manufacturer’s instructions. Total DNA content in PBMC lysates was measured spectrophotometrically using a GeneQuant device (Amersham Biosciences, Cambridge, UK). The results were expressed per µg of DNA.

### 4.3. Total RNA Isolation and Reverse Transcriptase (RT) Reaction

Total RNA was isolated from 100 µL of whole blood immediately after withdrawal using Extract RNA reagent (Evrogen, Moscow, Russia) following the manufacturer’s recommendations. The total RNA had an A260/290 ratio > 1.9. The RNA integrity number (RIN) was measured using G2991BA TapeStation 4200 (Agilent Technologies, Lexington, MA, USA). RNA samples with RIN values ≥ 8 were used for further analyses. The RT reaction was performed using an MMLV RT kit containing M-MLV Reverse Transcriptase, random hexanucleotide primers, and total RNA according to the manufacturer’s recommendations (Evrogen, Moscow, Russia).

### 4.4. Real-Time Quantitative PCR

The TaqMan primers and probes (Applied Biosystems, Foster City, CA, USA) for the expression of the human genes PKM2 (Hs00175407_m1), LDHB (Hs00929956_m1), SDHB (Hs01042482_m1), UCP2 (Hs01075227_m1), AMPKα (Hs01562315_m1), CPT1A (Hs00912671_m1), ACLY (Hs00982738_m1), IDH (Hs00188065_m1), MDH2 (Hs0093 8918_m1), PDH (Hs00264851_m1), ACACa (Hs01046047_m1), MLYCD (Hs00918031_m1), FASN (Hs01005622_m1), and ATP5B (Hs00969569_m1) were used. β-Actin served as an endogenous control.

We utilized standardized Pre-Developed TaqMan^®^ Gene Expression Assays from Applied Biosystems, which include pre-designed probe and primer sets for quantitative gene expression studies on human genes. These assays were validated by the supplier for specificity and efficiency. All Gene Expression Assays used in the study were designed to span an exon junction and exclude the detection of genomic DNA, eliminating the need for further quality assay analyses.

As an endogenous control we used only β-actin, which showed expression stability across our samples and an accurate normalization of gene expression data. In addition, β-actin was used in our previous studies, and its employment in this study was required for comparative analyses.

The quantification of gene expression was conducted using a Quant Studio 5 Real-Time PCR System (Applied Biosystems, Foster City, CA, USA). A volume of 1 μL of RT product was subjected to real-time PCR in a 15 μL total reaction mixture including 7.5 μL TaqMan Universal PCR Master Mix (Applied Biosystems), 900 nM sense and antisense primers, 50 nM probe, and a cDNA template. After a first step of 50 °C for 2 min and initial warming at 95 °C for 10 min, reaction mixtures were exposed to 40 amplification cycles (15 s at 95 °C for denaturation and 1 min of annealing and extension at 60 °C) [59]. Relative mRNA expression was determined using the delta-delta CT method, as described by the manufacturer (Applied Biosystems) [60].

### 4.5. Statistical Analysis

The data distribution was analyzed using the Shapiro–Wilk normality test. In the bivariate correlation analyses, Spearman’s rank correlation coefficients and the Mann–Whitney U-test were used. Receiver-operating characteristic (ROC) curve analyses were presented as the areas under the curve (AUCs) and the 95% confidence intervals (CIs). The diagnostic efficacies of the gene expression values were assessed using the sensitivities and specificities at the cut-off points. Each point on the ROC curve represents a sensitivity/specificity pair corresponding to a particular decision threshold. Therefore, for the location of the cut-off point for gene expression values, the middle ground between sensitivity and specificity was determined. In our case, there was no preference between sensitivity and specificity, and we used a reasonable approach to maximize both indices. The protein–protein interaction (PPI) network was created using the Search Tool for the Retrieval of Interacting Genes database (STRING, version 12.0) [61]. Statistical analyses were performed using Statistica for Windows (StatSoft Inc., version 10, Tulsa, OK, USA) and SPSS version 19 software (IBM, Armonk, NY, USA); *p*-values ≤ 0.05 were considered significant.

## 5. Conclusions

The end-stage patients with KOA who developed POP exhibited increased glycolytic pathway activity, disturbances in oxidative phosphorylation, and overexpression of both the CPT1A gene, responsible for the increased entry of long-chain fatty acids into mitochondria, and the ACLY gene, involved in Acetyl-CoA accumulation in the blood cells of patients before arthroplasty. Compared to pain-free individuals, patients who developed POP exhibited significantly increased energy requirements, as evidenced by elevated AMPK gene expression, suggesting enhanced intracellular biosynthetic activity, including the synthesis of proinflammatory cytokines. These findings establish a connection between pain and inflammation, which is driven by metabolic processes linked to energy production. Furthermore, the results suggest that the underlying and primary cause of OA involves disruptions in metabolic pathways that ultimately lead to the clinical symptoms of KOA. Therefore, our findings support our initial hypothesis proposing a relationship between carbohydrate and fatty acid metabolism and the development of postoperative pain (POP) in end-stage KOA patients, as well as identify potential molecular biomarkers for predicting POP development.

The ability to monitor the expression of these identified genes in peripheral blood allows for the early detection of metabolic abnormalities in end-stage KOA patients before surgery, enabling appropriate preventive measures to be taken. These results contribute to a better understanding of the metabolic processes involved in OA pathogenesis. Further research involving larger patient cohorts is necessary to validate our findings and strengthen our conclusions regarding the association between carbohydrate and fatty acid metabolism and POP development in end-stage KOA patients.

By analyzing gene expression patterns related to metabolic pathways and protein levels in PBMCs, we can gain a deeper understanding of the molecular mechanisms underlying individual patient responses to surgery and potentially predict their postoperative outcomes more accurately. This personalized approach based on gene expression profiling may offer valuable information for optimizing patient management strategies and improving surgical outcomes in the context of end-stage knee osteoarthritis.

## 6. Patients

A detailed description of the patients has been presented in our previous publication related to this patient cohort [9]. In brief, the study enrolled 50 unrelated subjects with primary OA of the knee, with an average age of 67.6 ± 7.5 years and a range of 54–82 years, who underwent primary total knee arthroplasty at the Nasonova Research Institute of Rheumatology between March 2018 and June 2019. Additionally, 26 healthy individuals of comparable age and gender (average age 65.8 ± 7.3 years, range 42–74 years) from the Moscow area were included as control subjects. All of the patients with end-stage OA fulfilled the criteria of the American College of Rheumatology regarding OA [62].

## Figures and Tables

**Figure 1 ijms-25-03857-f001:**
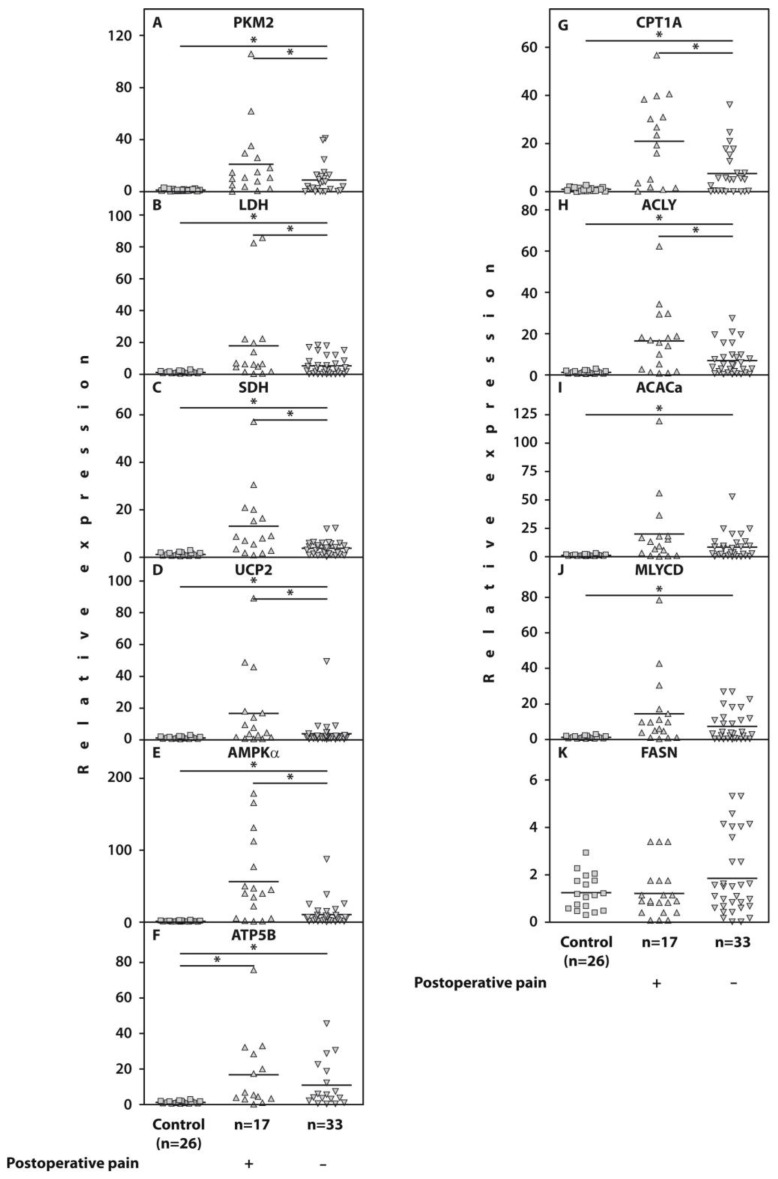
Relative expression of the genes PKM2 (**A**), LDH (**B**), SDH (**C**), UCP2 (**D**), AMPKα (**E**), ATP5B (**F**), CPT1A (**G**), ACLY (**H**) ACACa (**I**), MLYCD (**J**), and FASN (**K**) related to β-actin determined by real-time PCR analyses in the whole blood of end-stage KOA patients who either developed POP (n = 17) or were satisfied by surgery results (n = 33) compared to healthy controls (n = 26). Controls are shown as 1.0 as required for relative quantification with the real-time PCR protocol. Asterisks (*) indicate significant differences (Mann–Whitney U-test) between examined subsets of patients with end-stage KOA.

**Figure 2 ijms-25-03857-f002:**
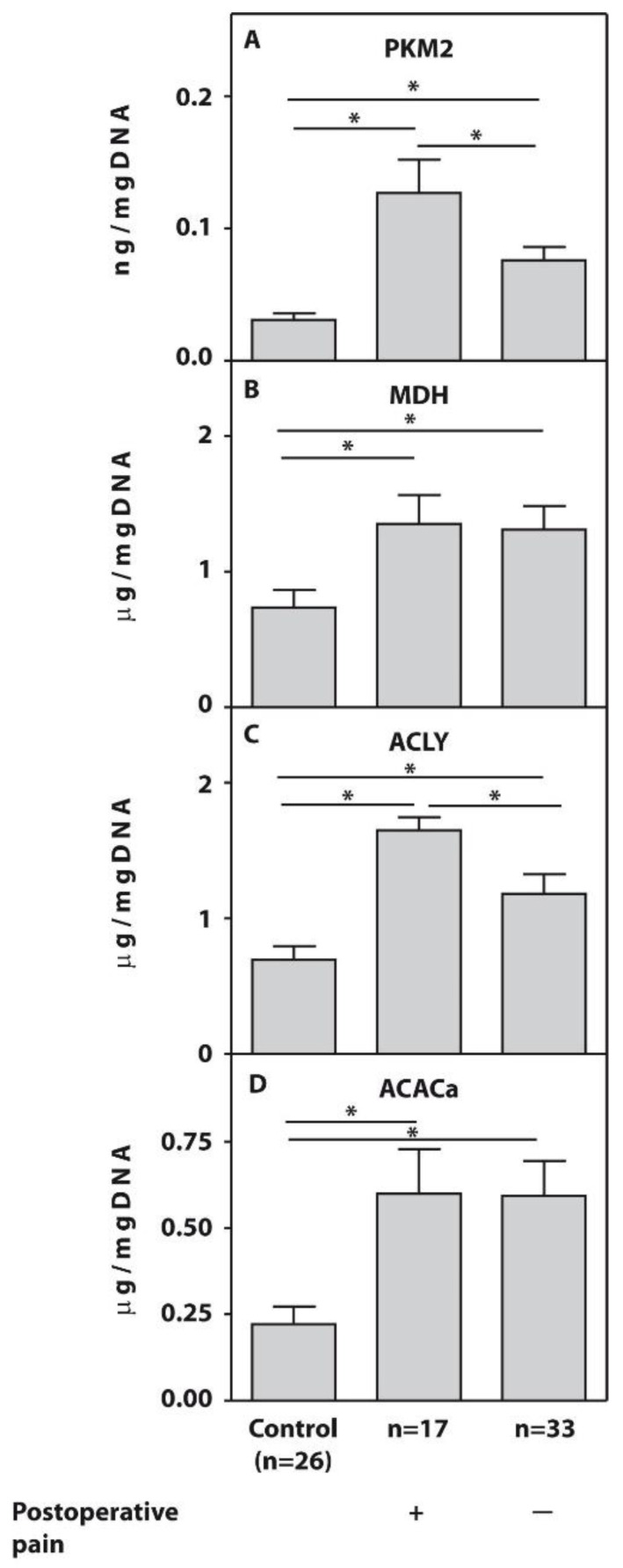
Protein concentrations of PKM2 (**A**), MDH (**B**), ACLY (**C**), and ACACa (**D**) measured by ELISA in PBMCs from patients with end-stage OA who developed POP (n = 17) and pain-free subjects (n = 33). Asterisk (*) indicates significant differences (Mann–Whitney U-test) between examined subsets.

**Figure 3 ijms-25-03857-f003:**
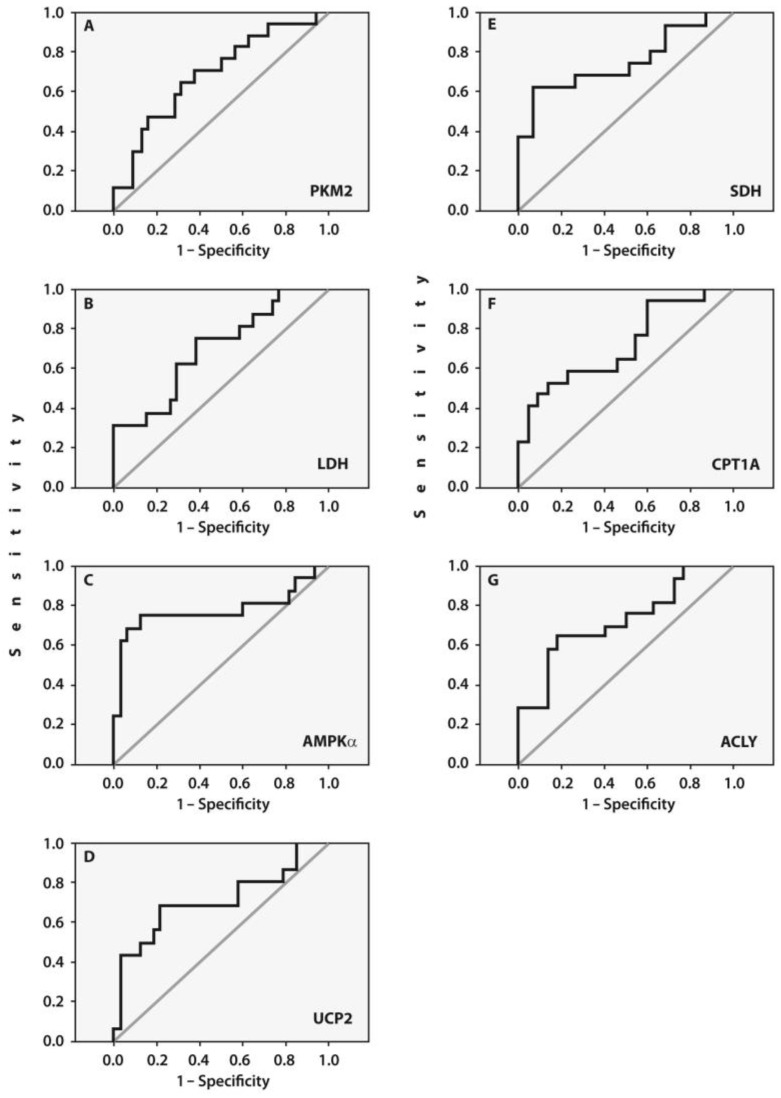
Areas under the curve (AUCs) between the baseline gene expressions in the peripheral blood of patients with end-stage OA who developed POP (n = 17) and pain-free subjects (n = 33) (**A**–**G**). Receiver-operating characteristic (ROC) curves for the expressions of (**A**) PKM2 [AUC = 0.689, 95%CI (0.532–0.846), *p* = 0.031], (**B**) LDH [AUC = 0.700, 95%CI (0.545–0.856), *p* = 0.023], (**C**) AMPKα [AUC = 0.777, 95%CI (0.605–0.950)], *p* = 0.002], (**D**) UCP2 [AUC = 0.716, 95%CI (0.546–0.885), *p* = 0.015], (**E**) SDH [AUC = 0.758, 95%CI (0.597–0.919), *p* = 0.004], (**F**) CPT1A [AUC = 0.719, 95%CI (0.588–0.773), *p* = 0.02], (**G**) ACLY [AUC = 0.727, 95%CI (0.563–0.891), *p* = 0.01].

**Figure 4 ijms-25-03857-f004:**
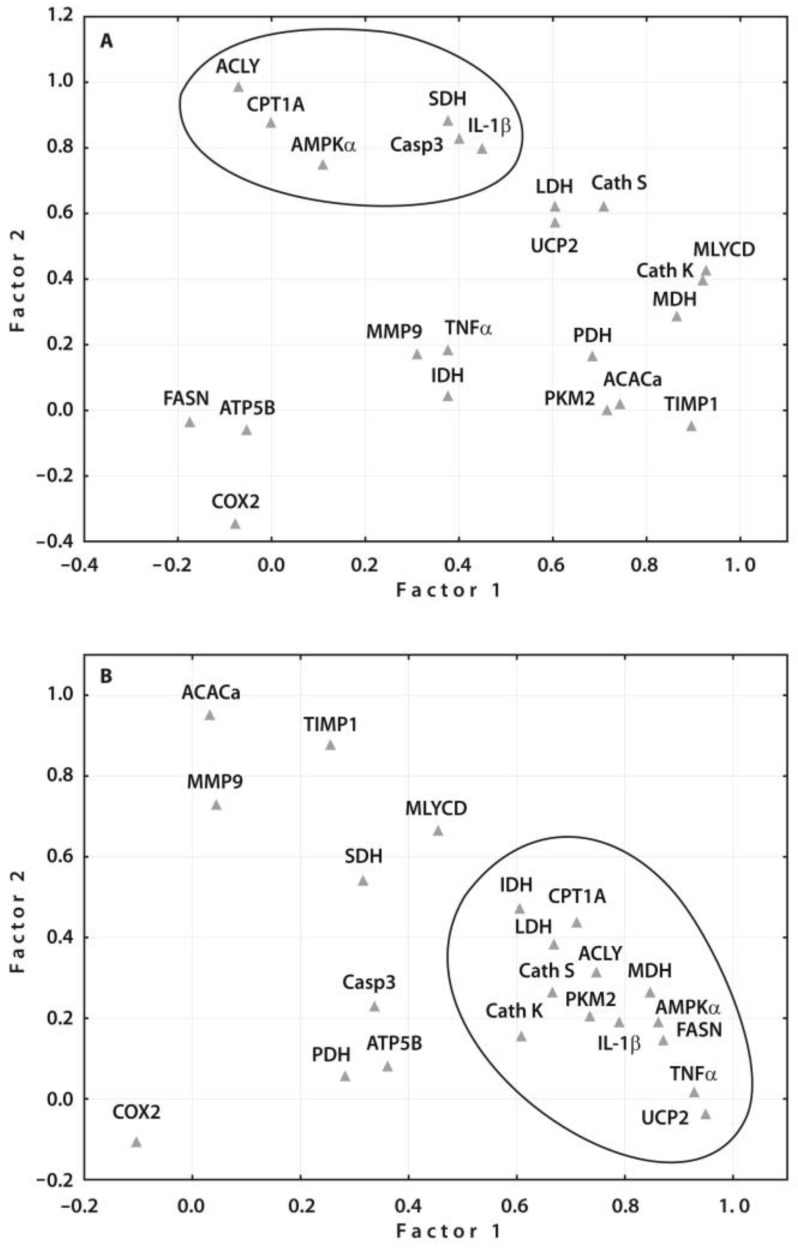
Subgroups of the examined gene expressions (n = 22) following principal component analysis. (**A**) The factor analysis suggested one main factor containing 6 genes in patients with KOA who developed POP and (**B**) another main factor containing 13 genes in pain-free patients after surgery.

**Figure 5 ijms-25-03857-f005:**
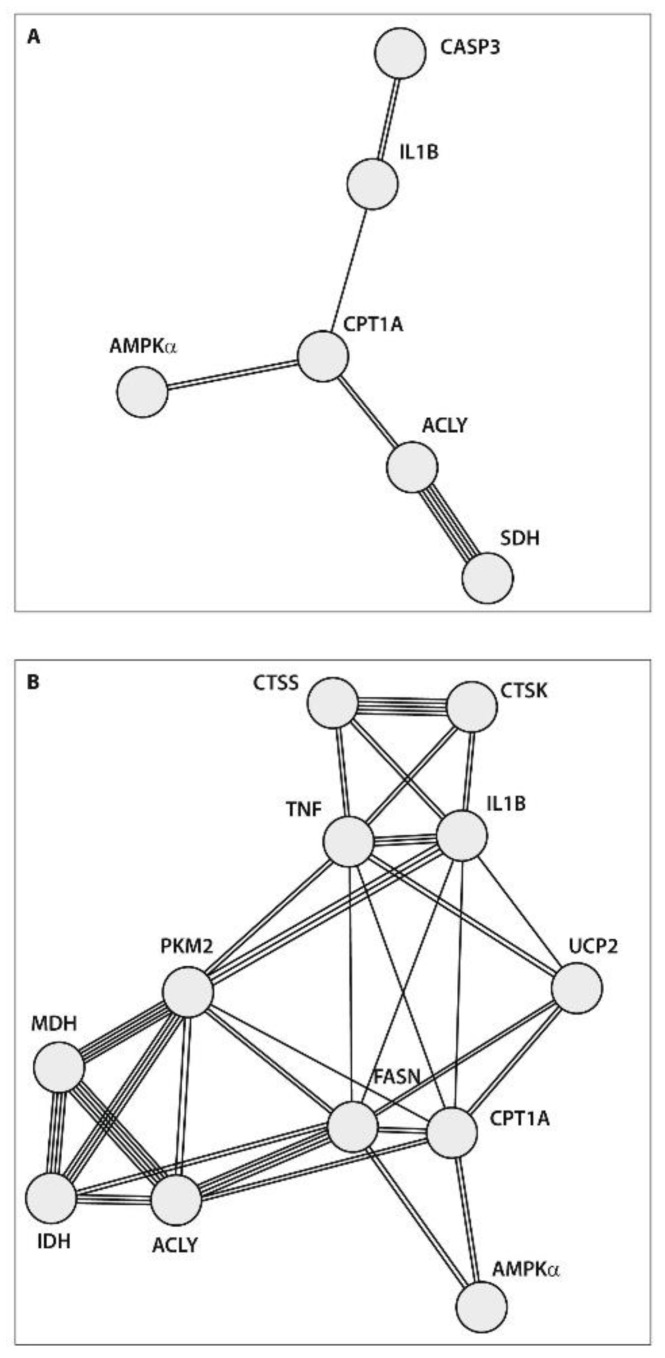
Protein–protein interactions of the examined gene expressions prior to surgery in the peripheral blood of end-stage patients with knee OA: (**A**) patients who developed POP, (**B**) pain-free subjects.

**Table 1 ijms-25-03857-t001:** Relative expression of the genes PKM2, LDH, SDH, MDH, IDH, UCP2, AMPKα, ATP5B, CPT1A, ACLY, ACACa, MLYCD, and FASN related to β-actin determined by real-time PCR analyses in the whole blood of end-stage KOA patients who either developed POP (n = 17) or were satisfied by surgery results (n = 33). ns, non-significant.

Gene	Patients Who Developed POP n = 17	Pain-Free Subjects n = 33	*p* (Mann–Whitney U-Test)
PKM2	12.6 [5.7; 27.8]	4.1 [0.3; 11.5]	0.03
LDH	6.4 [3.2; 20.9]	3.9 [0.9; 7.6]	0.04
SDH	8.3 [2.3; 20.5]	3.7 [1.0; 5.7]	0.02
MDH	5.8 [1.4; 30.6]	3.2 [0.6; 5.5]	ns
IDH	4.8 [2.1; 18.3]	4.8 [2.1; 8.5]	ns
UCP2	6.1 [1.6; 17.4]	1.8 [0.8; 2.8]	0.02
AMPKα	39.9 [3.6; 122.0]	6.4 [1.6; 16.8]	0.001
ATP5B	6.1 [3.2; 30.4]	4.8 [1.7; 20.6]	ns
CPT1A	21.7 [2.9; 35.0]	6.0 [0.9; 17.0]	0.01
ACLY	16.2 [3.9; 24.0]	5.0 [1.7; 9.1]	0.01
ACACa	13.2 [3.2; 18.2]	4.0 [1.3; 12.9]	ns
MLYCD	9.8 [3.8; 17.1]	3.9 [0.7; 13.2]	ns
FASN	1.1 [0.7; 3.0]	1.5 [0.7; 2.8]	ns

**Table 2 ijms-25-03857-t002:** Spearman’s rank correlation coefficients and their significance (*p*) between clinical indices and gene expressions measured before surgery in patients with end-stage KOA (n = 50).

	AMPKα	PKM2	UCP2	ACLY	MLYCD	CPT1A
DN4	0.423 *p* = 0.013	0.471 *p* = 0.004	0.33 *p* = 0.046	0.446 *p* = 0.006	0.368 *p* = 0.03	0.368 *p* = 0.027
PainDETECT		0.343 *p* = 0.063	0.342 *p* = 0.06	0.456 *p* = 0.011	0.459 *p* = 0.012	0.388 *p* = 0.037

## Data Availability

The data presented in this study are available on request from the corresponding author.

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
