# Peer review of "Metabolic Dysregulation and Its Role in Postoperative Pain among Knee Osteoarthritis Patients"

_ijms, 2024, doi:10.3390/ijms25073857_

Round 1
Reviewer 1 Report
Comments and Suggestions for Authors
Dear Authors,
I was pleased to review the paper entitled "Importance of dysregulation in carbohydrate and fatty acid metabolism for the development of postoperative pain in end-stage patients with knee osteoarthritis" -
- MDPI –
The present paper is very interesting, it focuses on a relevant clinical scenario, for orthopedics, potentially influencing the surgical and clinical practice for the management of knee disease.
Therefore, it is my opinion that the content is original, current, and relevant.
Thus, there are some minor remarks:
- Title: The title gives a fine idea of the topic to be covered.
- Abstract: lines 18-19, better explain your idea: the pain may resist even post knee replacement until when?
- Introduction:
Well written, clearly describes the study objectives and background.
This section is too long and risks losing the reader's attention. I suggest focusing on the biomedical data concerning the object of analysis.
- Method: The method section is postponed in the paper. I suggest moving as section no2.
- Result: well written.
Would a possible multivariate analysis (logistic regression) yield statistically significant benefits? for example, are depression or other humor disorders as important as metabolic alterations?
- Discussion:
One limitation of the analysis is the postoperative evaluation of the replacement implants. There are several causes of post-op pain such as malalignment, aseptic loosening or excessive bleeding (you could add doi: 10.52965/001c.37625). You would have to show that all implants have been validated by orthopedic surgeons and have no inherent abnormalities, so the cause of pain may come from something else.
End the section with limits, strengths and future implication of the research.
The paper generally is well written.
Comments on the Quality of English LanguageMinor editing of English language required
Author Response
RESPONSES TO REVIEWER 1
We are grateful to the Reviewer for detailed analysis of the manuscript. Changes in the text are highlighted.
- Abstract: lines 18-19, better explain your idea: the pain may resist even post knee replacement until when?
This has been corrected.
We modified Lines 18-19: Strong, continuous pain may indicate the need for joint replacement in patients with end-stage OA although postoperative pain (POP) of at least two months duration persists in 10-40% of patients with OA.
In addition, we put in some information on chronic pain criteria and its possible duration to the Introduction section. Lines (61-66):
The specific criteria for chronic postoperative pain were defined recently [7]: The pain must develop after a surgical procedure; The pain is of at least two months duration; Other causes for the pain have been excluded; The possibility that the pain is from a pre-existing condition has been excluded. Chronic POP can last rather long as it was observed in patients up to 12 years after surgery [8].
- Introduction:
This section is too long and risks losing the reader's attention. I suggest focusing on the biomedical data concerning the object of analysis.
This has been corrected. We removed several sentences from the Introduction section and focused more directly on the novel aspects of the current study.
We also added on Lines (69-78):
Local articular cartilage degradation, changes in subchondral bone, and synovial inflammation have long been considered the major characteristics of KOA [1,2]. However, recently, several systemic molecular mechanisms have been implicated in the pathogenesis and progression of the disease. These mechanisms involve chronic low-grade inflammation due to the upregulation of proinflammatory cytokines, which promote the upregulation of matrix metalloproteinases, aggrecanases, and cathepsins, leading to the breakdown of the cartilage extracellular matrix. These activities are accompanied by dysregulation of metabolic pathways, such as glucose and lipid metabolism, and mitochondrial dysfunction, which can affect chondrocyte homeostasis and contribute to cartilage degeneration [10].
However, as we have to justify the involvement of all the examined genes, we preserved descriptions of the molecular pathways associated with the tested metabolic conversions.
- Method: The method section is postponed in the paper. I suggest moving as section no2.
We placed Methods section after Discussion as it is a Requirement of the International Journal of Molecular Science.
Would a possible multivariate analysis (logistic regression) yield statistically significant benefits? for example, are depression or other humor disorders as important as metabolic alterations?
The reliability of these cut-off values was confirmed by logistic regression modeling, which showed that high expression of AMPKα (p=0.0025), SDH (p=0.03), CPT1A (p=0.03), and ACLY (p=0.028) were independent predictors of POP development. However, these results warrant further investigation due to the limited number of patients in both subsets.
Lines (271-280)
However, logistic regression analyses of clinical traits and comorbidities in the same subsets of patients did not show any significant differences, probably due to the small size of our patient cohorts. However, in our future studies involving significantly higher numbers of patients with OA undergoing knee arthroplasty these analyses would have to be conducted.
- Discussion:
One limitation of the analysis is the postoperative evaluation of the replacement implants. There are several causes of post-op pain such as malalignment, aseptic loosening or excessive bleeding (you could add doi: 10.52965/001c.37625). You would have to show that all implants have been validated by orthopedic surgeons and have no inherent abnormalities, so the cause of pain may come from something else.
This has been corrected. We added this information to the section of study strengths. Lines (27-30; 418-424):
All implants have been validated by orthopedic surgeons, and patients with OA demonstrated no inherent abnormalities to cause pain from other reasons than OA disease such as malalignment, aseptic loosening, or excessive bleeding. Indeed, patients with vascular insufficiency, bleeding, or thrombophlebitis were excluded from our study as indicated in our previous report within the same patient cohort [12]. In addition, intravenous tranexamic acid was applied before the skin incision to prevent postoperative bleeding after arthroplasty [12,51].
End the section with limits, strengths and future implication of the research.
This has been corrected. We added on Lines (418-475):
Our study had several strengths. First, we carefully selected the patients with OA who did not have inherent abnormalities that could cause pain from reasons other than OA, such as malalignment, aseptic loosening, or excessive bleeding. Additionally, patients with vascular insufficiency, bleeding, or thrombophlebitis were excluded from the study as previously indicated in our report within the same patient cohort [12]. Furthermore, intravenous tranexamic acid was administered before the skin incision to prevent postoperative bleeding after arthroplasty [12,51]. Second, all implants were validated by orthopedic surgeons. Third, to explore and support our hypothesis, we utilized various research approaches, including examination of gene expression and cellular protein levels, correlation and ROC curve analyses, PCA, and PPI network construction. All these approaches yielded similar results. However, our study also had a limitation in that it involved a small number of patients. This limitation may have contributed to non-significant results in some of our tests.
Based on our findings, we propose several areas for future research. For example, further investigation into other metabolic pathways that may be dysregulated in osteoarthritis could provide a more comprehensive understanding of the molecular mechanisms underlying POP development. Additionally, developing targeted therapies based on the identified metabolic disturbances in OA could lead to more effective treatments for the disease prior to surgery. Furthermore, validation studies involving larger patient cohorts are needed to confirm the findings of our study and ensure the reproducibility of the results. Overall, these future research directions have the potential to advance our understanding of the molecular mechanisms underlying osteoarthritis and pave the way for the development of novel therapeutic strategies to improve patient care.
Our study brings several novel contributions to the field of osteoarthritis (OA) and postoperative pain research. We have identified specific metabolic disturbances associated with KOA that have not been extensively studied before. By uncovering dysregulation in key metabolic pathways, our study sheds light on the underlying molecular mechanisms contributing to KOA pathogenesis. Additionally, our findings establish a link between metabolic disturbances and postoperative pain outcomes in KOA patients. This novel insight highlights the potential role of metabolism in modulating pain sensitivity and underscores the importance of considering metabolic factors in pain management strategies for KOA patients. By pinpointing metabolic pathways that are dysregulated in KOA, our study opens up new avenues for developing targeted therapies to address these metabolic disturbances. This has significant implications for the development of personalized treatment approaches that target specific metabolic pathways to alleviate symptoms and improve patient outcomes. Overall, our study contributes to advancing the understanding of KOA by elucidating the intricate interplay between metabolism and pain.
Our findings not only contribute to the understanding of osteoarthritis (OA) at the molecular level but also have broader implications for public health and socioeconomic factors. As OA is a leading cause of disability worldwide, affecting millions of individuals and placing a significant burden on healthcare systems, by uncovering metabolic disturbances associated with KOA and their link to POP outcomes, our study provides valuable insights that can inform public health strategies for accurate management of OA. The identification of dysregulation in energy metabolic pathways in KOA opens up possibilities for personalized medicine approaches tailored to individual patients' metabolic profiles. This precision medicine approach has the potential to optimize treatment outcomes, reduce healthcare costs, and improve patient satisfaction and quality of life. KOA is not only a major health issue but also carries significant socioeconomic implications due to its impact on work productivity, healthcare expenditures, and quality of life. By shedding light on the metabolic factors influencing pain outcomes in KOA patients, our study may lead to more cost-effective treatment strategies and improved outcomes, thereby reducing the economic burden associated with OA. Finally, our study sets the stage for further research into the role of metabolism in OA pathogenesis and pain modulation. Future studies could explore the potential of targeting specific metabolic pathways for therapeutic interventions, as well as investigate the impact of lifestyle factors, such as diet and exercise, on metabolic disturbances in OA. By placing our findings in the larger context of OA research, public health implications, and socioeconomic impacts, we underscore the significance of our work in advancing knowledge, improving patient care, and addressing the multifaceted challenges posed by this prevalent musculoskeletal condition.
Comments on the Quality of English Language Minor editing of English language required
This has been corrected.

Reviewer 2 Report
Comments and Suggestions for Authors
For Title and Abstract:
Title Brevity and Clarity: The title is overly long and could be made more concise. Consider revising to focus on the core of the study, such as "Metabolic Dysregulation and Its Role in Postoperative Pain among Knee Osteoarthritis Patients".
Abstract Structure: The abstract should succinctly present the study's purpose, methods, key findings, and conclusions. It appears to cover these elements but could benefit from clearer demarcation and brevity in each section to enhance readability and immediate understanding.
Specificity in the Introduction: The introduction to the condition (KOA) is broad. Specifying how dysregulation in carbohydrate and fatty acid metabolism specifically contributes to KOA and postoperative pain early in the abstract would help focus the reader's understanding.
Methodology Details: While the abstract mentions the collection of peripheral blood samples and the assessment of pain levels and functional activity, it could benefit from a brief explanation of the selection criteria for the study population and any controls for variables that could affect the outcomes.
Results Clarity: The results are presented with significant findings but could be improved by briefly mentioning the statistical significance of the gene expressions observed (e.g., p-values) to substantiate the claims of significant differences between groups.
Discussion of Implications: The abstract concludes with the implications of the findings but could further highlight the novel contributions of this study to the field. Specifically, elucidating how these findings could influence future research, clinical practice, or both would be beneficial.
Keywords Optimization: The keywords are appropriate but might be refined to include broader terms that encompass the study's scope, such as "metabolic dysregulation" or "arthroplasty outcomes", to improve searchability and reach a wider audience.
Consistency in Terminology: Ensure consistent use of terms throughout the abstract. For instance, if "postoperative pain (POP)" is defined, it should consistently be referred to as such throughout the text.
Technical Details: The method of analysis (quantitative real-time RT-PCR, ELISA) is mentioned, but specifying why these particular methods were chosen over others could add depth to the methodological understanding.
For Introduction:
Streamline Introduction: While comprehensive, the introduction could be streamlined for clarity and impact. Consider condensing background information to focus more directly on the novel aspects of the current study.
Clarify Research Gap: Explicitly state the gap in the current research landscape that this study aims to fill. Doing so will highlight the contribution of your work to the field.
Objective Statement Precision: The statement of purpose is somewhat buried in the text. Make the study's objectives clear and succinct, ideally in a standalone sentence early in the introduction.
Enhance Cohesion: Some paragraphs cover a broad range of topics. Consider reorganizing content to ensure that each paragraph focuses on a single main idea, enhancing readability and logical flow.
Cite Recent Studies: Where possible, cite recent studies to support your statements, especially when discussing the prevalence of OA and the effectiveness of current treatments. This ensures the relevance and timeliness of your background information.
Specific Hypothesis Statement: The hypothesis is mentioned towards the end but could be more precisely stated. Clearly articulate the expected relationships or outcomes your study seeks to verify.
Technical Terms and Acronyms: Ensure that all technical terms and acronyms are clearly defined the first time they are used. This includes terms like PBMCs, which may not be familiar to all readers.
Background on KOA: Expand on the molecular mechanisms of KOA more comprehensively to establish a stronger foundation for understanding the significance of your study.
Tighten Language: Some sentences are overly complex or lengthy. Aim for concise, direct sentences that convey your points more clearly to improve readability.
Bridge to Methods: End the introduction with a smooth transition that connects the background and objectives to the methodology of the study. This helps readers understand how you plan to address the research question.
For Results:
Clarification of Clinical Parameters: The explanation of clinical parameters and their relevance could be more clearly connected to the study's objectives. Specifically, elaborate on how these parameters directly relate to the hypothesis being tested.
Statistical Analysis Description: The text should more explicitly describe the statistical methods used to analyze the data. Including statistical tests, significance levels (e.g., p-values), and confidence intervals provides clarity and allows readers to assess the robustness of the findings.
Interpretation of WOMAC Scores: While the average total WOMAC score is presented, a brief explanation of what these scores represent in the context of KOA severity would aid reader understanding, especially for those less familiar with the scale.
Graphs and Figures Reference: The results reference several figures (e.g., Figure 1, Figure 2) without providing descriptions or interpretations of these visuals within the text. Summarizing key findings from each figure in the text would enhance comprehension for readers not viewing the figures directly.
Clear Differentiation of Subgroups: The distinction between patients with and without postoperative pain needs to be clearer, especially in how their clinical and molecular profiles compare and contrast. This differentiation is crucial for understanding the study's conclusions.
Presentation of Gene Expression Data: When discussing gene expression differences, the text should more systematically present these findings, perhaps in a table format within the article, to succinctly convey which genes were upregulated or not significantly different.
Protein Level Findings: The significance of the protein level findings in the context of the study's hypothesis should be more thoroughly discussed. Explain how these findings contribute to understanding postoperative pain mechanisms in KOA.
Correlation Analysis Interpretation: While the correlations between gene expressions and clinical parameters are mentioned, a discussion on the implications of these correlations for KOA pathology and treatment outcomes would provide valuable insight.
ROC Curve Analysis: The presentation of ROC curve analysis results should include a discussion on the clinical relevance of the identified cut-off values and how they might be used in predicting postoperative pain.
Technical Terms Explanation: Ensure that all technical terms and acronyms (e.g., PKM2, LDH, PPI) are clearly defined when first mentioned to make the text accessible to a wider audience.
PPI Network Analysis: The implications of the PPI network analysis findings for understanding the molecular interactions involved in KOA and postoperative pain should be discussed more comprehensively.
For Discussion:
Clarify the Link Between Bioenergetics and Inflammation: While the text mentions the interaction between cellular bioenergetics/metabolism and inflammation, it could benefit from a more detailed explanation of how these processes interconnect in the context of KOA. Providing more direct evidence or theories would strengthen this link.
Enhance Comparative Analysis: When comparing your results with previous studies or animal models, ensure a thorough analysis that not only highlights similarities but also explores potential reasons for discrepancies. This could involve discussing methodological differences or the complexity of translating findings from animal models to human disease.
Discuss Non-Significant Findings: The discussion mentions no significant differences in certain gene expressions. Delve deeper into why these results might have occurred and their implications. Discussing non-significant findings can provide insight into the condition's complexity and suggest directions for future research.
Mechanistic Insights: While the text makes connections between gene expressions and POP, it often stops short of fully explaining the mechanisms at play. Expanding on how these gene expressions contribute to the biochemical pathways leading to POP would add depth to the discussion.
Contextualize Gene Expression Findings: The significant increase in certain gene expressions is mentioned, but the discussion could benefit from contextualizing these findings within the broader landscape of OA research. How do these findings shift or affirm our understanding of OA pathophysiology?
Address Contradictions or Limitations: If there are contradictions within the findings or compared to other studies, address them directly. Discussing limitations in your study's design or data interpretation can provide a balanced view and suggest areas for improvement in future research.
Implications for Treatment or Diagnosis: The discussion should extend the findings' implications for clinical practice, such as potential targets for therapeutic intervention or diagnostic markers. Suggestions for how these findings might influence patient management or treatment options would be valuable.
Future Research Directions: Propose specific areas for future research based on your findings. This could include exploring additional molecular pathways, conducting longitudinal studies to track gene expression changes over time, or developing targeted therapies based on the identified metabolic disturbances.
Integration with Existing Literature: While some references to previous studies are made, integrating your findings more comprehensively with the existing body of knowledge will strengthen the discussion. This includes comparing your results with a broader range of studies and meta-analyses in the field.
Clearer Explanation of Technical Terms and Processes: Ensure that all technical terms and biological processes are clearly explained for readers who may not be specialists in this particular area of research. Simplifying complex concepts without losing accuracy can make the discussion more accessible.
Highlight Novelty and Contribution: Explicitly state what new knowledge your study brings to the field. Emphasize the unique contributions and how they advance our understanding of KOA and POP.
Broader Contextualization: Discuss how your findings fit into the larger context of OA research, public health implications, and potential socioeconomic impacts. This broader perspective can underline the significance of your work.
For Materials and Methods:
Standardize Sample Collection Information: Specify the rationale for collecting blood samples between 07:00 AM and 09:00 AM. If the timing is related to potential diurnal variations in gene expression or protein levels, this should be clearly stated.
Clarify Sample Processing: When describing the process of isolating peripheral blood mononuclear cells (PBMCs), provide more detail on the Ficoll density gradient used, including specific concentrations or gradients, to allow for reproducibility.
Detail ELISA Procedure: While ELISA kit references are provided, including specific conditions such as incubation times, temperatures, and any steps taken to ensure accuracy and reliability of the measurements could enhance methodological clarity.
RNA Isolation and Quality: Mention any steps taken to ensure the integrity and purity of RNA beyond the A260/290 ratio, such as RNA integrity number (RIN) evaluation or gel electrophoresis images, which could reinforce the quality of the starting material for RT reactions.
PCR Conditions and Validation: Detail the validation of the primers and probes used, including efficiency curves, to confirm their specificity and efficiency. This adds to the reliability of the quantitative PCR (qPCR) results.
Normalization and Control in qPCR: Expand on the choice of β-actin as an endogenous control. Discuss its expression stability across your samples or whether other housekeeping genes were considered to ensure accurate normalization of gene expression data.
Statistical Methods Detail: While statistical tests are mentioned, elaborating on the choice of tests for specific types of data and explaining any transformations or adjustments made (e.g., Bonferroni correction for multiple comparisons) would strengthen the statistical analysis section.
ROC Curve Analysis Clarification: Detail the process for determining cut-off points for gene expression values. Explain how these were chosen and justify their relevance to the study's objectives.
PPI Network Analysis Details: Provide more information on the criteria for selecting genes for the PPI network analysis, the version of the STRING database used, and any thresholds for interaction confidence scores to ensure transparency.
Software Versions and Settings: Mention specific versions of the statistical software used and any relevant settings or packages. This information is crucial for reproducibility.
Ethical Approval and Consent: Confirm that ethical approval was obtained for the study and that patients provided informed consent for their participation. This is a critical aspect of research involving human subjects
Data Availability Statement: Consider including a statement on the availability of the data generated from the study. This could involve detailing how and where the data can be accessed, aligning with open science practices.
For Conclusions:
Clarity and Specificity: The conclusions could be more precise in summarizing the main findings. For instance, specifying how the increased glycolytic pathway activity and disturbances in oxidative phosphorylation directly relate to postoperative pain could clarify the study's implications.
Connection to Hypotheses: Reinforce how the findings support or refute the initial hypotheses presented in the introduction. This ties the article together and provides a satisfying conclusion to the reader.
Implications for Clinical Practice: Briefly discuss the potential implications of these findings for clinical practice. For example, mention if the identified metabolic disturbances could lead to new biomarkers for predicting postoperative pain or targets for therapeutic intervention.
Limitations and Future Directions: While the need for further studies is acknowledged, expanding on specific limitations of the current study (e.g., sample size, study design) and proposing detailed future research directions could strengthen the conclusions.
Broader Context: Place the study's findings within the broader context of OA research. Highlighting how this study adds to the current understanding of OA pathogenesis could elevate its perceived importance.
Comments on the Quality of English Language
Language and Grammar Corrections
Incorrect Use of Hyphenation:
Original: "Importance of dysregulation in carbohydrate and fatty acid me- tabolism..."
Corrected: "Importance of dysregulation in carbohydrate and fatty acid metabolism..."
Inconsistent Line Breaks:
Original text contains improper line breaks (e.g., "me- tabolism").
Correction involves removing unnecessary line breaks to ensure continuity in text.
Grammatical Errors:
Original: "aiming to suppress of inflammation and pain."
Corrected: "aiming to suppress inflammation and pain."
Ambiguous Pronoun Reference:
Original: "These findings suggest that disturbances in energy metabolic conversions observed in PBMCs of end-stage patients with KOA before arthroplasty may contribute to the development of POP."
Corrected: "These findings suggest that the disturbances in energy metabolism, as observed in the PBMCs of patients with end-stage KOA before arthroplasty, may contribute to POP development."
Unclear Statistical Presentation:
Original: "patients who developed postoperative pain suffered more from grade 1 obesity (p = 0.08) and anxiety according to the HADS questionnaire (p = 0.07)."
Corrected: "A higher incidence of grade 1 obesity (p = 0.08) and elevated anxiety levels, as measured by the HADS questionnaire (p = 0.07), were observed in patients who developed postoperative pain."
Scientific Expression Enhancements
Vague Descriptions:
Original: "The proliferation of immune cells and synthesis of proinflammatory cytokines are energy-consuming process..."
Corrected: "The proliferation of immune cells and the synthesis of proinflammatory cytokines constitute energy-intensive processes..."
Misleading Representation of Data:
Original: "patients (n=17) who reported POP showed significantly higher gene expression..."
Improved Clarity: "Among the study cohort, 17 patients who reported POP demonstrated significantly higher expressions of genes..."
Lack of Specificity in Methods:
Original: "Peripheral blood (10 ml) was collected in vacutainers containing ethylenediaminetetraacetic acid (EDTA)."
More Detail Needed: "We collected 10 ml of peripheral blood in vacutainers that contained ethylenediaminetetraacetic acid (EDTA) to prevent coagulation, ensuring the integrity of the blood samples for subsequent analyses."
Ambiguous Conclusions:
Original: "These patients experienced a significantly greater en- ergy requirements compared to pain-free patients..."
Precise Language: "Compared to pain-free individuals, patients who developed POP exhibited significantly increased energy requirements, as evidenced by elevated AMPK gene expression, suggesting enhanced intracellular biosynthetic activity."
Author Response
RESPONSES TO REVIEWER 2
We are grateful to the Reviewer for detailed analysis of the manuscript. Changes in the text are highlighted.
For Title and Abstract:
Title Brevity and Clarity: The title is overly long and could be made more concise. Consider revising to focus on the core of the study, such as "Metabolic Dysregulation and
Its Role in Postoperative Pain among Knee Osteoarthritis Patients".
This has been corrected.
Abstract Structure: The abstract should succinctly present the study's purpose, methods, key findings, and conclusions. It appears to cover these elements but could benefit from clearer demarcation and brevity in each section to enhance readability and immediate understanding.
This has been corrected.
Specificity in the Introduction: The introduction to the condition (KOA) is broad. Specifying how dysregulation in carbohydrate and fatty acid metabolism specifically contributes to KOA and postoperative pain early in the abstract would help focus the reader's understanding.
This has been corrected. We added to the Abstract Lines (21-24):
The inflammation observed in joint tissues is linked to pain caused by the production of proinflammatory cytokines. Since the biosynthesis of cytokines requires energy, their production is supported by extensive metabolic conversions of carbohydrates and fatty acids, which could lead to a disruption of cellular homeostasis.
Methodology Details: While the abstract mentions the collection of peripheral blood samples and the assessment of pain levels and functional activity, it could benefit from a brief explanation of the selection criteria for the study population and any controls for variables that could affect the outcomes.
This has been corrected. We added to the Abstract Lines (27-30): All implants have been validated by orthopedic surgeons and patients with OA demonstrated no inherent abnormalities to cause pain from other reasons than OA disease such as malalignment, aseptic loosening or excessive bleeding.
Results Clarity: The results are presented with significant findings but could be improved by briefly mentioning the statistical significance of the gene expressions observed (e.g., p-values) to substantiate the claims of significant differences between groups.
This has been corrected. We added P<0.05 to indicate statistical significance of the gene expressions.
Discussion of Implications: The abstract concludes with the implications of the findings but could further highlight the novel contributions of this study to the field. Specifically, elucidating how these findings could influence future research, clinical practice, or both would be beneficial.
This has been corrected. We added We added on Line (47-48): Additionally, our findings can be used in a clinical setting to predict POP development in end-stage patients with KOA before arthroplasty.
Keywords Optimization: The keywords are appropriate but might be refined to include broader terms that encompass the study's scope, such as "metabolic dysregulation" or "arthroplasty outcomes", to improve searchability and reach a wider audience.
This has been corrected. We added additional keywords: “metabolic dysregulation" and "arthroplasty outcomes".
Consistency in Terminology: Ensure consistent use of terms throughout the abstract. For instance, if "postoperative pain (POP)" is defined, it should consistently be referred to as such throughout the text.
This has been corrected.
Technical Details: The method of analysis (quantitative real-time RT-PCR, ELISA) is mentioned, but specifying why these particular methods were chosen over others could add depth to the methodological understanding.
This has been corrected. We used qRT-PCR because it is the most sensitive and reliable method for gene expression analysis while ELISA was used to confirm our qRT-PCR results.
For Introduction:
Streamline Introduction: While comprehensive, the introduction could be streamlined for clarity and impact. Consider condensing background information to focus more directly on the novel aspects of the current study.
This has been corrected. We removed several sentences from the Introduction section and focused more directly on the novel aspects of the current study. Lines (117-124):
Previous studies investigating pain in KOA primarily focused on inflammatory mechanisms. However, analyzing proinflammatory cytokine production alone may not uncover the underlying causes of pain development. Therefore, we chose to explore another aspect of cellular vitality related to general energy production mechanisms as a potential primary factor influencing metabolic fluctuations.
Clarify Research Gap: Explicitly state the gap in the current research landscape that this study aims to fill. Doing so will highlight the contribution of your work to the field.
This has been corrected. We added on Lines (117-124):
Previous studies investigating pain in KOA primarily focused on inflammatory mechanisms. However, analyzing proinflammatory cytokine production alone may not uncover the underlying causes of pain development. Therefore, we chose to explore another aspect of cellular vitality related to general energy production mechanisms as a potential primary factor influencing metabolic fluctuations.
Objective Statement Precision: The statement of purpose is somewhat buried in the text. Make the study's objectives clear and succinct, ideally in a standalone sentence early in the introduction.
This has been corrected. We placed the purpose of the study in a separate paragraph. Lines (125-128).
Enhance Cohesion: Some paragraphs cover a broad range of topics. Consider reorganizing content to ensure that each paragraph focuses on a single main idea, enhancing readability and logical flow.
This has been corrected. We removed several sentences from the Introduction section and focused more directly on the novel aspects of the current study
Cite Recent Studies: Where possible, cite recent studies to support your statements, especially when discussing the prevalence of OA and the effectiveness of current treatments. This ensures the relevance and timeliness of your background information.
This has been corrected. We substituted references ##5,6,7 for:
- Katz JN, Arant KR, Loeser RF. Diagnosis and treatment of hip and knee osteoarthritis: a review. JAMA. 2021;325(6):568–78. https://doi.org/10. 1001/jama.2020.22171
- Feng H, Feng ML, Cheng JB, Zhang X, Tao HC. Meta-analysis of factors influencing anterior kneepainafter total knee arthroplasty. World J Orthop. 2024 Feb 18;15(2):180-191. doi: 10.5312/wjo.v15.i2.180.
- Simpson AHRW, Clement ND, Simpson SA, Pandit H, Smillie S, Leeds AR, Conaghan PG, Kingsbury SR, Hamilton D, Craig P, et al. A preoperativepackage of care for osteoarthritis, consisting of weight loss, orthotics, rehabilitation, and topical and oral analgesia (OPPORTUNITY): a two-centre, open-label, randomised controlled feasibility trial. J. Lancet Rheumatol. 2024 Feb 26:S2665-9913(23)00337-5. doi: 10.1016/S2665-9913(23)00337-5.
Specific Hypothesis Statement: The hypothesis is mentioned towards the end but could be more precisely stated. Clearly articulate the expected relationships or outcomes your study seeks to verify.
This has been corrected. We rephrased our hypothesis as follows:
Here we hypothesized that POP development in KOA might be associated with overall disturbances in carbohydrate and fatty acid metabolism that can be monitored in peripheral blood mononuclear cells (PBMCs). Lines (121-124)
Technical Terms and Acronyms: Ensure that all technical terms and acronyms are clearly defined the first time they are used. This includes terms like PBMCs, which may not be familiar to all readers.
This has been corrected.
Background on KOA: Expand on the molecular mechanisms of KOA more comprehensively to establish a stronger foundation for understanding the significance of your study.
This has been corrected. We added on Lines (69-78):
Local articular cartilage degradation, changes in subchondral bone, and synovial inflammation have long been considered the major characteristics of KOA [REF]. However, recently, several systemic molecular mechanisms have been implicated in the pathogenesis and progression of the disease. These mechanisms involve chronic low-grade inflammation due to the upregulation of proinflammatory cytokines, which promote the upregulation of matrix metalloproteinases, aggrecanases, and cathepsins, leading to the breakdown of the cartilage extracellular matrix. These activities are accompanied by dysregulation of metabolic pathways, such as glucose and lipid metabolism, and mitochondrial dysfunction, which can affect chondrocyte homeostasis and contribute to cartilage degeneration [REF].
We added on Lines (117-124):
Previous studies investigating pain in KOA primarily focused on inflammatory mechanisms. However, analyzing proinflammatory cytokine production alone may not uncover the underlying causes of pain development. Therefore, we chose to explore another aspect of cellular vitality related to general energy production mechanisms as a potential primary factor influencing metabolic fluctuations. Here we hypothesized that POP development in KOA might be associated with overall disturbances in carbohydrate and fatty acid metabolism that can be monitored in peripheral blood mononuclear cells (PBMCs).
Tighten Language: Some sentences are overly complex or lengthy. Aim for concise, direct sentences that convey your points more clearly to improve readability.
This has been corrected.
Bridge to Methods: End the introduction with a smooth transition that connects the background and objectives to the methodology of the study. This helps readers understand how you plan to address the research question.
This has been corrected. We added on Lines (129-134):
In light of the issues mentioned above, we employed various approaches to analyze gene and protein expressions associated with energy production in the peripheral blood of end-stage KOA patients before surgery. These approaches included correlation studies, ROC curve analyses, logistic regression modeling, principal component analyses, and construction of protein-protein interaction networks to identify and validate genes with significant prognostic power.
For Results:
Clarification of Clinical Parameters: The explanation of clinical parameters and their relevance could be more clearly connected to the study's objectives. Specifically, elaborate on how these parameters directly relate to the hypothesis being tested.
This has been corrected. We added on Lines (170-174):
However, these differences in the manifestation of the examined clinical parameters of the end-stage KOA patients could not help to distinguish patients who will develop POP or not. In addition, clinical traits cannot provide any indications on the intrinsic mechanisms of POP development. Therefore, we used molecular approaches aiming to solve the problem indicated as the purpose of our study.
Statistical Analysis Description: The text should more explicitly describe the statistical methods used to analyze the data. Including statistical tests, significance levels (e.g., p-values), and confidence intervals provides clarity and allows readers to assess the robustness of the findings.
This has been corrected. We added methods used to analyze the data, including statistical tests, significance levels (e.g., p-values), and confidence intervals where they were missing.
Interpretation of WOMAC Scores: While the average total WOMAC score is presented, a brief explanation of what these scores represent in the context of KOA severity would aid reader understanding, especially for those less familiar with the scale.
This has been corrected. We added on Lines (145-148): The Western Ontario and McMaster Universities Arthritis Index (WOMAC) is widely used in the evaluation of Knee Osteoarthritis. It is a self-administered questionnaire consisting of 24 items divided into three subscales: pain, stiffness, and physical function.
Graphs and Figures Reference: The results reference several figures (e.g., Figure 1, Figure 2) without providing descriptions or interpretations of these visuals within the text. Summarizing key findings from each figure in the text would enhance comprehension for readers not viewing the figures directly.
This has been corrected. We added on Lines (184-190):
Figure 1 consists of two panels. The left panel presents the expressions of genes related to glycolysis (PKM2, LDH), the Krebs cycle (SDH), electron transport chain uncoupling protein (UCP2), ATP synthase (ATP5B), and AMP-activated protein kinase (AMPKα), the major regulator of glucose and fatty acid metabolism [28]. The right panel introduces the expressions of genes related to long-chain fatty acid β-oxidation (CPT1A) and fatty acid synthesis (ACLY, ACACa, MLYCD, and FASN) [29].
In Figure 2, we demonstrate that the obtained gene expression results align with protein expression data obtained after analyzing protein samples from PBMCs of the same patients using ELISA. These analyses revealed significant differences in gene expressions of PKM2 and ACLY between the examined subgroups of end-stage KOA patients, supported by significant differences in protein concentrations within the same subgroups. Conversely, the lack of significant differences in the gene expression of MDH and ACACa resulted in no significant difference in their cellular protein levels.
Clear Differentiation of Subgroups: The distinction between patients with and without postoperative pain needs to be clearer, especially in how their clinical and molecular profiles compare and contrast. This differentiation is crucial for understanding the study's conclusions.
This has been corrected. We added on Lines (127-130):
The comparison of outcomes from analyses of clinical and molecular profiles suggests that gene expression examination is a powerful tool for differentiating patients prior to arthroplasty. This approach not only helps identify patients at risk of postoperative pain (POP) but also provides insights into the potential reasons for POP development.
Presentation of Gene Expression Data: When discussing gene expression differences, the text should more systematically present these findings, perhaps in a table format within the article, to succinctly convey which genes were upregulated or not significantly different.
This has been corrected. We added Table 1 on Lines (191-196):
Table 1. Relative expression of the genes PKM2, LDH, SDH, MDH, IDH, UCP2, AMPKα, ATP5B, CPT1A, ACLY, ACACa, MLYCD, and FASN related to β-actin determined by real-time PCR analyses in the whole blood of end-stage KOA patients who either developed POP (n=17) or were satisfied by surgery results (n=33).
|
Gene |
Patients who developed POP n=17 |
Pain-free subjects n=33 |
p (Mann-Whitney U-test) |
|
PKM2 |
12.6 [5.7; 27.8] |
4.1 [0.3; 11.5] |
0.03 |
|
LDH |
6.4 [3.2; 20.9] |
3.9 [0.9; 7.6] |
0.04 |
|
SDH |
8.3 [2.3; 20.5] |
3.7 [1.0; 5.7] |
0.02 |
|
MDH |
5.8 [1.4; 30.6] |
3.2[0.6; 5.5] |
ns |
|
IDH |
4.8 [ 2.1; 18.3] |
4.8 [2.1; 8.5] |
ns |
|
UCP2 |
6.1 [1.6; 17.4] |
1.8 [0.8; 2.8] |
0.02 |
|
AMPKα |
39.9 [3.6;122.0] |
6.4 [1.6;16.8] |
0.001 |
|
ATP5B |
6.1 [3.2;30.4] |
4.8 [1.7;20.6] |
ns |
|
CPT1A |
21.7 [2.9; 35.0] |
6.0 [0.9; 17.0] |
0.01 |
|
ACLY |
16.2 [3.9; 24.0] |
5.0 [1.7; 9.1] |
0.01 |
|
ACACa |
13.2 [3.2; 18.2] |
4.0 [1.3; 12.9] |
ns |
|
MLYCD |
9.8 [3.8; 17.1] |
3.9 [0.7; 13.2] |
ns |
|
FASN |
1.1 [0.7; 3.0] |
1.5 [0.7; 2.8] |
ns |
Protein Level Findings: The significance of the protein level findings in the context of the study's hypothesis should be more thoroughly discussed. Explain how these findings contribute to understanding postoperative pain mechanisms in KOA.
This has been corrected. We added on Lines (220-231):
In Figure 2, we demonstrate that the obtained gene expression results align with protein expression data obtained after analyzing protein samples from PBMCs of the same patients using ELISA. These analyses revealed significant differences in gene expressions of PKM2 and ACLY between the examined subgroups of end-stage KOA patients, supported by significant differences in protein concentrations within the same subgroups. Conversely, the lack of significant differences in the gene expression of MDH and ACACa resulted in no significant difference in their cellular protein levels.
The comparison of outcomes from analyses of clinical and molecular profiles suggests that gene expression examination is a powerful tool for differentiating patients prior to arthroplasty. This approach not only helps identify patients at risk of postoperative pain (POP) but also provides insights into the potential reasons for POP development.
Correlation Analysis Interpretation: While the correlations between gene expressions and clinical parameters are mentioned, a discussion on the implications of these correlations for KOA pathology and treatment outcomes would provide valuable insight.
This has been corrected. We added on Lines (244-249):
The correlation analyses revealed that BMI is associated with respiration efficacy (UCP2) and de novo fatty acid synthesis (MLYCD). Neuropathic pain, as indicated by DN4 and Pain DETECT scores, is linked to the overall regulation of cellular energy metabolism (AMPKα), as well as both the beta-oxidation of long-chain fatty acids (CPT1A) and their de novo synthesis (ACLY, MLYCD). Additionally, pain severity increases when Krebs cycle activity decreases (MDH).
ROC Curve Analysis: The presentation of ROC curve analysis results should include a discussion on the clinical relevance of the identified cut-off values and how they might be used in predicting postoperative pain.
This has been corrected. We added on Lines (273-280):
The above-mentioned cut-off values obtained in ROC curve analyses allow for the use of gene expression data measured in peripheral blood in end-stage KOA patients for predicting postoperative pain (POP) before surgery. Specifically, gene expression values exceeding the cut-off value indicate a higher likelihood of POP development.
The reliability of these cut-off values was confirmed by logistic regression modeling, which showed that high expression of AMPKα (p=0.0025), SDH (p=0.03), CPT1A (p=0.03), and ACLY (p=0.028) were independent predictors of POP development. However, these results warrant further investigation due to the limited number of patients in both subsets.
Technical Terms Explanation: Ensure that all technical terms and acronyms (e.g., PKM2, LDH, PPI) are clearly defined when first mentioned to make the text accessible to a wider audience.
This has been corrected.
PPI Network Analysis: The implications of the PPI network analysis findings for understanding the molecular interactions involved in KOA and postoperative pain should be discussed more comprehensively.
This has been corrected. We added on Lines (293-297):
Furthermore, PCA results supported the prognostic significance of the same gene expressions identified through ROC curve analyses and logistic regression modeling. Nevertheless, PCA analysis of gene expressions in pain-free KOA patients revealed a greater number of interconnected genes, indicating stronger regulatory mechanisms associated with less severe disease (Figure 4B).
We added on Lines (306-315):
Protein-protein interactions (PPIs) represent specific physical contacts between two or more protein molecules driven by biochemical events involving electrostatic forces, hydrogen bonding, and hydrophobic effects. The degree of connectivity in PPI networks is determined by the number of links to a given node. In the present study, PPI network analysis showed that end-stage patients who developed POP exhibited a strong association between ACLY and SDH genes, which are linked to fatty acid synthesis and respiratory activities, respectively (Figure 5A). At the same time, the degree of connectivity and the number of genes involved were stronger in patients who were satisfied with surgery results indicating more powerful regulatory connections between various metabolic branches in these subjects (Figure 5B).
For Discussion:
Clarify the Link Between Bioenergetics and Inflammation: While the text mentions the interaction between cellular bioenergetics/metabolism and inflammation, it could benefit from a more detailed explanation of how these processes interconnect in the context of KOA. Providing more direct evidence or theories would strengthen this link.
This has been corrected. We added on Lines (326-333):
Indeed, excessive proinflammatory cytokine synthesis in inflamed tissues requires glucose breakdown through aerobic glycolysis to convert pyruvate into lactate (known as the Warburg effect) [11]. This mechanism allows for the rapid generation of high amounts of energy in the form of ATP. Simultaneously, the anti-inflammatory activities of oxidative mitochondrial metabolism are suppressed [14]. In healthy subjects, acute inflammation is followed by the restoration of energy balance with a predominance of Krebs cycle activity. However, during chronic inflammation in KOA patients, these mechanisms could be dysregulated.
Enhance Comparative Analysis: When comparing your results with previous studies or animal models, ensure a thorough analysis that not only highlights similarities but also explores potential reasons for discrepancies. This could involve discussing methodological differences or the complexity of translating findings from animal models to human disease.
This has been corrected. We added on Lines (359-363):
It is worth noting that, although animal studies support molecular findings in humans, end-stage KOA disease affects older patients, making it challenging to accurately model in animal studies. Additionally, OA disease in animals is artificially induced by known agents, while the molecular triggers of human OA are currently unclear. Therefore, results from animal model studies should always be verified by human research.
Discuss Non-Significant Findings: The discussion mentions no significant differences in certain gene expressions. Delve deeper into why these results might have occurred and their implications. Discussing non-significant findings can provide insight into the condition's complexity and suggest directions for future research.
This has been corrected. We added on Lines (348-350):
The absence of significant changes in the expression of genes related to oxidative mitochondrial metabolism support previous observations of Krebs cycle downregulation compared to glycolysis activity associated with inflammation [17].
Mechanistic Insights: While the text makes connections between gene expressions and POP, it often stops short of fully explaining the mechanisms at play. Expanding on how these gene expressions contribute to the biochemical pathways leading to POP would add depth to the discussion.
This has been corrected. We added on Lines (339-342):
Mechanistically, inflammation fueled by upregulation of glycolysis leads to the release of inflammatory mediators, which can bind to their respective receptors on nociceptive neurons in the periphery, modulating their sensitivity and excitability, resulting in increased pain sensitivity [35].
Contextualize Gene Expression Findings: The significant increase in certain gene expressions is mentioned, but the discussion could benefit from contextualizing these findings within the broader landscape of OA research. How do these findings shift or affirm our understanding of OA pathophysiology?
This has been corrected. We added on Lines (396-406):
Overall, our study demonstrated that chronic POP is associated with significant metabolic dysfunction involving carbohydrate and fatty acid metabolism in end-stage patients with KOA. As pain in the knee joint is considered a key trait related to OA disease, we can suggest that similar metabolic disturbances may occur in patients with OA at other disease stages. Moreover, as we monitored these changes in peripheral blood, metabolic dysfunction may involve other tissues and organs of the body. Therefore, OA might not be a disease exclusive to the joint but rather a systemic condition involving the entire body, and knee arthroplasty may not represent the end stage of OA disease.
Address Contradictions or Limitations: If there are contradictions within the findings or compared to other studies, address them directly. Discussing limitations in your study's design or data interpretation can provide a balanced view and suggest areas for improvement in future research.
This has been corrected. We added on Lines (418-430):
Our study had several strengths. First, we carefully selected the patients with OA who did not have inherent abnormalities that could cause pain from reasons other than OA, such as malalignment, aseptic loosening, or excessive bleeding. Additionally, patients with vascular insufficiency, bleeding, or thrombophlebitis were excluded from the study as previously indicated in our report within the same patient cohort [12]. Furthermore, intravenous tranexamic acid was administered before the skin incision to prevent postoperative bleeding after arthroplasty [12,51]. Second, all implants were validated by orthopedic surgeons. Third, to explore and support our hypothesis, we utilized various research approaches, including examination of gene expression and cellular protein levels, correlation and ROC curve analyses, PCA, and PPI network construction. All these approaches yielded similar results. However, our study also had a limitation in that it involved a small number of patients. This limitation may have contributed to non-significant results in some of our tests.
Implications for Treatment or Diagnosis: The discussion should extend the findings' implications for clinical practice, such as potential targets for therapeutic intervention or diagnostic markers. Suggestions for how these findings might influence patient management or treatment options would be valuable.
This has been corrected. We added on Lines (407-417):
Therefore, our findings suggest new approaches for POP diagnostics and treatment. We have identified a number of prognostic biomarkers for POP development that can be measured in peripheral blood before arthroplasty. This means that end-stage KOA patients at risk of POP development may receive treatment before surgery to target the improvement of their metabolic condition. In the future, new approaches for potential therapies could be applied to prevent POP development in end-stage KOA patients by addressing intrinsic disturbances in cellular metabolism.
Future Research Directions: Propose specific areas for future research based on your findings. This could include exploring additional molecular pathways, conducting longitudinal studies to track gene expression changes over time, or developing targeted therapies based on the identified metabolic disturbances.
This has been corrected. We added on Lines (431-440):
Based on our findings, we propose several areas for future research. For example, further investigation into other metabolic pathways that may be dysregulated in osteoarthritis could provide a more comprehensive understanding of the molecular mechanisms underlying POP development. Additionally, developing targeted therapies based on the identified metabolic disturbances in OA could lead to more effective treatments for the disease prior to surgery. Furthermore, validation studies involving larger patient cohorts are needed to confirm the findings of our study and ensure the reproducibility of the results. Overall, these future research directions have the potential to advance our understanding of the molecular mechanisms underlying osteoarthritis and pave the way for the development of novel therapeutic strategies to improve patient care.
Integration with Existing Literature: While some references to previous studies are made, integrating your findings more comprehensively with the existing body of knowledge will strengthen the discussion. This includes comparing your results with a broader range of studies and meta-analyses in the field.
This has been corrected. We added following references:
48.Genetic Links Between Metabolic Syndrome and Osteoarthritis: Insights from Cross-Trait Analysis. Huang JX, Xu SZ, Tian T, Wang J, Jiang LQ, He T, Meng SY, Ni J, Pan HF.J Clin Endocrinol Metab. 2024 Mar 14:dgae169. doi: 10.1210/clinem/dgae169. Online ahead of print.
49.Factors associated with pain and functional impairment five years after total knee arthroplasty: a prospective observational study. Olsen U, Sellevold VB, Gay CL, Aamodt A, Lerdal A, Hagen M, Dihle A, Lindberg MF.BMC Musculoskelet Disord. 2024 Jan 2;25(1):22. doi: 10.1186/s12891-023-07125-y.
50.Enhanced recovery after surgery in patients after hip and knee arthroplasty: a systematic review and meta-analysis. Zhang Q, Chen Y, Li Y, Liu R, Rai S, Li J, Hong P.Postgrad Med J. 2024 Feb 15;100(1181):159-173. doi: 10.1093/postmj/qgad125.
Clearer Explanation of Technical Terms and Processes: Ensure that all technical terms and biological processes are clearly explained for readers who may not be specialists in this particular area of research. Simplifying complex concepts without losing accuracy can make the discussion more accessible.
This has been corrected.
Highlight Novelty and Contribution: Explicitly state what new knowledge your study brings to the field. Emphasize the unique contributions and how they advance our understanding of KOA and POP.
This has been corrected. We added on Lines (441-453):
Our study brings several novel contributions to the field of osteoarthritis (OA) and postoperative pain research. We have identified specific metabolic disturbances associated with KOA that have not been extensively studied before. By uncovering dysregulation in key metabolic pathways, our study sheds light on the underlying molecular mechanisms contributing to KOA pathogenesis. Additionally, our findings establish a link between metabolic disturbances and postoperative pain outcomes in KOA patients. This novel insight highlights the potential role of metabolism in modulating pain sensitivity and underscores the importance of considering metabolic factors in pain management strategies for KOA patients. By pinpointing metabolic pathways that are dysregulated in KOA, our study opens up new avenues for developing targeted therapies to address these metabolic disturbances. This has significant implications for the development of personalized treatment approaches that target specific metabolic pathways to alleviate symptoms and improve patient outcomes. Overall, our study contributes to advancing the understanding of KOA by elucidating the intricate interplay between metabolism and pain.
Broader Contextualization: Discuss how your findings fit into the larger context of OA research, public health implications, and potential socioeconomic impacts. This broader perspective can underline the significance of your work.
This has been corrected. We added on Lines (454-475):
Our findings not only contribute to the understanding of osteoarthritis (OA) at the molecular level but also have broader implications for public health and socioeconomic factors. As OA is a leading cause of disability worldwide, affecting millions of individuals and placing a significant burden on healthcare systems, by uncovering metabolic disturbances associated with KOA and their link to POP outcomes, our study provides valuable insights that can inform public health strategies for accurate management of OA. The identification of dysregulation in energy metabolic pathways in KOA opens up possibilities for personalized medicine approaches tailored to individual patients' metabolic profiles. This precision medicine approach has the potential to optimize treatment outcomes, reduce healthcare costs, and improve patient satisfaction and quality of life. KOA is not only a major health issue but also carries significant socioeconomic implications due to its impact on work productivity, healthcare expenditures, and quality of life. By shedding light on the metabolic factors influencing pain outcomes in KOA patients, our study may lead to more cost-effective treatment strategies and improved outcomes, thereby reducing the economic burden associated with OA. Finally, our study sets the stage for further research into the role of metabolism in OA pathogenesis and pain modulation. Future studies could explore the potential of targeting specific metabolic pathways for therapeutic interventions, as well as investigate the impact of lifestyle factors, such as diet and exercise, on metabolic disturbances in OA. By placing our findings in the larger context of OA research, public health implications, and socioeconomic impacts, we underscore the significance of our work in advancing knowledge, improving patient care, and addressing the multifaceted challenges posed by this prevalent musculoskeletal condition.
For Materials and Methods:
Standardize Sample Collection Information: Specify the rationale for collecting blood samples between 07:00 AM and 09:00 AM. If the timing is related to potential diurnal variations in gene expression or protein levels, this should be clearly stated.
This has been corrected. We added on Line (497): after overnight fasting and before breakfast.
Clarify Sample Processing: When describing the process of isolating peripheral blood mononuclear cells (PBMCs), provide more detail on the Ficoll density gradient used, including specific concentrations or gradients, to allow for reproducibility.
This has been corrected. We added on Lines (498-507):Whole blood was separated using a Ficoll density gradient. PBMCs, which have a lower density than Ficoll-Hypaque (1.077 g/L), were separated from higher density granulocytes and red blood cells through centrifugation after overlaying the diluted blood on the Ficoll-Hypaque layer. Following centrifugation, the blood samples were divided into plasma enriched with thrombocytes, PBMCs located in the interphase, and a pellet containing granulocytes above the red blood cells. The PBMCs from the interphase were isolated and washed twice in phosphate-buffered saline. Erythrocytes were lysed using a hypotonic buffer (1.6 mM EDTA, 10 mM KHCO3, 153 mM NH4Cl, pH 7.4) at a 3:1 volume ratio. The isolated PBMCs were then frozen and stored at -80°C until protein extraction.
Detail ELISA Procedure: While ELISA kit references are provided, including specific conditions such as incubation times, temperatures, and any steps taken to ensure accuracy and reliability of the measurements could enhance methodological clarity.
This has been corrected. We presented the general Protocol of ELISA. Lines (512-518): Standard solutions were provided in the kit. Diluted samples and standards were added to the appropriate wells of the pre-coated 96-well plate. After incubating for 1 hour at 37°C, a detection reagent was added, followed by another 1-hour incubation at 37°C, a washing procedure, and additional incubation with a second detection reagent for 30 minutes at 37°C. Following another washing process, the plate was incubated with a substrate solution for 30 minutes at 37°C, then a stop solution was added, and the plate was immediately read using a microplate reader at 450nm.
RNA Isolation and Quality: Mention any steps taken to ensure the integrity and purity of RNA beyond the A260/290 ratio, such as RNA integrity number (RIN) evaluation or gel electrophoresis images, which could reinforce the quality of the starting material for RT reactions.
This has been corrected. We added on Lines (530-532): RNA integrity number (RIN) was measured using G2991BA TapeStation 4200 (Agilent Technologies). RNA samples with RIN values ≥ 8 were used for further analyses.
PCR Conditions and Validation: Detail the validation of the primers and probes used, including efficiency curves, to confirm their specificity and efficiency. This adds to the reliability of the quantitative PCR (qPCR) results.
This has been corrected. We added on Lines (544-549): We utilized standardized Pre-Developed TaqMan® Gene Expression Assays from Applied Biosystems, which include pre-designed probe and primer sets for quantitative gene expression studies on human genes. These assays were validated by the supplier for specificity and efficiency. All Gene Expression Assays used in the study were designed to span an exon junction and exclude detection of genomic DNA, eliminating the need for further quality assay analyses.
Normalization and Control in qPCR: Expand on the choice of β-actin as an endogenous control. Discuss its expression stability across your samples or whether other housekeeping genes were considered to ensure accurate normalization of gene expression data.
This has been corrected. We added on Lines (550-553): As an endogenous control we used only β-actin which showed expression stability across our samples and accurate normalization of gene expression data. In addition, β-actin was used in our previous studies and its employment in this study was required for comparative analyses.
Statistical Methods Detail: While statistical tests are mentioned, elaborating on the choice of tests for specific types of data and explaining any transformations or adjustments made (e.g., Bonferroni correction for multiple comparisons) would strengthen the statistical analysis section.
We designed our study for the analyses of two subsets of end-stage KOA patients related to POP development. As we do not conduct multiple comparisons, we have no need for Bonferroni correction for multiple comparisons.
ROC Curve Analysis Clarification: Detail the process for determining cut-off points for gene expression values. Explain how these were chosen and justify their relevance to the study's objectives.
This has been corrected. We added on Lines (571-575):
Each point on the ROC curve represents a sensitivity/specificity pair corresponding to a particular decision threshold. Therefore, for location of the cut-off point for gene expression values the middle ground between sensitivity and specificity was determined. As in our case there was no preference between sensitivity and specificity, we used a reasonable approach to maximize both indices.
PPI Network Analysis Details: Provide more information on the criteria for selecting genes for the PPI network analysis, the version of the STRING database used, and any thresholds for interaction confidence scores to ensure transparency.
As it was stated in the previous version of the manuscript, for PPI Network Analysis we used all the gene expression data that was acquired in the present study and the previous study [12] conducted with the same subsets of patients. We used STRING database version 12.0 in our study. Line (577).
Software Versions and Settings: Mention specific versions of the statistical software used and any relevant settings or packages. This information is crucial for reproducibility.
This has been corrected. Lines (577-579): We used Statistica for Windows (StatSoft Inc., version 10, Tulsa, OK, USA) and SPSS version 19 software (IBM, Armonk, NY, USA).
Ethical Approval and Consent: Confirm that ethical approval was obtained for the study and that patients provided informed consent for their participation. This is a critical aspect of research involving human subjects
Institutional Review Board Statement has been already presented in the previous version of the paper. Lines (632-633): The study protocol was approved by the Local Committee on the Ethics of Human Research (Protocol No. 32 from December 20, 2018).
Data Availability Statement: Consider including a statement on the availability of the data generated from the study. This could involve detailing how and where the data can be accessed, aligning with open science practices.
This has been corrected. We added on Line (637-638):
The data presented in this study are available on request from the corresponding author.
For Conclusions:
Clarity and Specificity: The conclusions could be more precise in summarizing the main findings. For instance, specifying how the increased glycolytic pathway activity and disturbances in oxidative phosphorylation directly relate to postoperative pain could clarify the study's implications.
This has been corrected. We added on Lines (589-592):
These findings establish a connection between pain and inflammation, which is driven by metabolic processes linked to energy production. Furthermore, the results suggest that the underlying and primary cause of OA involves disruptions in metabolic pathways that ultimately lead to the clinical symptoms of KOA.
Connection to Hypotheses: Reinforce how the findings support or refute the initial hypotheses presented in the introduction. This ties the article together and provides a satisfying conclusion to the reader.
This has been corrected. We added on Lines (593-596):
Therefore, our findings support our initial hypothesis proposing a relationship between carbohydrate and fatty acid metabolism and the development of postoperative pain (POP) in end-stage KOA patients, as well as identify potential molecular biomarkers for predicting POP development.
Implications for Clinical Practice: Briefly discuss the potential implications of these findings for clinical practice. For example, mention if the identified metabolic disturbances could lead to new biomarkers for predicting postoperative pain or targets for therapeutic intervention.
This has been corrected. We added on Lines (597-599):
The ability to monitor the expression of these identified genes in peripheral blood allows for the early detection of metabolic abnormalities in end-stage KOA patients before surgery, enabling appropriate preventive measures to be taken.
Limitations and Future Directions: While the need for further studies is acknowledged, expanding on specific limitations of the current study (e.g., sample size, study design) and proposing detailed future research directions could strengthen the conclusions.
This has been corrected. We added on Lines (600-603):
Further research involving larger patient cohorts is necessary to validate our findings and strengthen our conclusions regarding the association between carbohydrate and fatty acid metabolism and POP development in end-stage KOA patients.
Broader Context: Place the study's findings within the broader context of OA research. Highlighting how this study adds to the current understanding of OA pathogenesis could elevate its perceived importance.
This has been corrected. We added on Lines (604-609):
By analyzing gene expression patterns related to metabolic pathways and protein levels in PBMCs, we can gain a deeper understanding of the molecular mechanisms underlying individual patient responses to surgery and potentially predict their postoperative outcomes more accurately. This personalized approach based on gene expression profiling may offer valuable information for optimizing patient management strategies and improving surgical outcomes in the context of end-stage knee osteoarthritis.
Comments on the Quality of English Language
Language and Grammar Corrections
This has been corrected (blue highlights).
Incorrect Use of Hyphenation:
Original: "Importance of dysregulation in carbohydrate and fatty acid me- tabolism..."
Corrected: "Importance of dysregulation in carbohydrate and fatty acid metabolism..."
Inconsistent Line Breaks:
Original text contains improper line breaks (e.g., "me- tabolism").
Correction involves removing unnecessary line breaks to ensure continuity in text.
Grammatical Errors:
Original: "aiming to suppress of inflammation and pain."
Corrected: "aiming to suppress inflammation and pain."
Ambiguous Pronoun Reference:
Original: "These findings suggest that disturbances in energy metabolic conversions observed in PBMCs of end-stage patients with KOA before arthroplasty may contribute to the development of POP."
Corrected: "These findings suggest that the disturbances in energy metabolism, as observed in the PBMCs of patients with end-stage KOA before arthroplasty, may contribute to POP development."
Unclear Statistical Presentation:
Original: "patients who developed postoperative pain suffered more from grade 1 obesity (p = 0.08) and anxiety according to the HADS questionnaire (p = 0.07)."
Corrected: "A higher incidence of grade 1 obesity (p = 0.08) and elevated anxiety levels, as measured by the HADS questionnaire (p = 0.07), were observed in patients who developed postoperative pain."
Scientific Expression Enhancements
Vague Descriptions:
Original: "The proliferation of immune cells and synthesis of proinflammatory cytokines are energy-consuming process..."
Corrected: "The proliferation of immune cells and the synthesis of proinflammatory cytokines constitute energy-intensive processes..."
Misleading Representation of Data:
Original: "patients (n=17) who reported POP showed significantly higher gene expression..."
Improved Clarity: "Among the study cohort, 17 patients who reported POP demonstrated significantly higher expressions of genes..."
Lack of Specificity in Methods:
Original: "Peripheral blood (10 ml) was collected in vacutainers containing ethylenediaminetetraacetic acid (EDTA)."
More Detail Needed: "We collected 10 ml of peripheral blood in vacutainers that contained ethylenediaminetetraacetic acid (EDTA) to prevent coagulation, ensuring the integrity of the blood samples for subsequent analyses."
Ambiguous Conclusions:
Original: "These patients experienced a significantly greater energy requirements compared to pain-free patients..."
Precise Language: "Compared to pain-free individuals, patients who developed POP exhibited significantly increased energy requirements, as evidenced by elevated AMPK gene expression, suggesting enhanced intracellular biosynthetic activity."
